# Extraction of active RhoGTPases by RhoGDI regulates spatiotemporal patterning of RhoGTPases

Adriana E Golding[1†], Ilaria Visco[2†], Peter Bieling[2]*, William M Bement[3,4]*

[1]Graduate Program in Cell and Molecular Biology, University of Wisconsin, Madison, United States; [2]Department of Systemic Cell Biology, Max Planck Institute of Molecular Physiology, Dortmund, Germany; [3]Laboratory of Cell and Molecular Biology, University of Wisconsin-Madison, Madison, United States; [4]Department of Integrative Biology, University of Wisconsin-Madison, Madison, United States

**Abstract** The RhoGTPases are characterized as membrane-associated molecular switches that cycle between active, GTP-bound and inactive, GDP-bound states. However, 90–95% of RhoGTPases are maintained in a soluble form by RhoGDI, which is generally viewed as a passive shuttle for inactive RhoGTPases. Our current understanding of RhoGTPase:RhoGDI dynamics has been limited by two experimental challenges: direct visualization of the RhoGTPases in vivo and reconstitution of the cycle in vitro. We developed methods to directly image vertebrate RhoGTPases in vivo or on lipid bilayers in vitro. Using these methods, we identified pools of active and inactive RhoGTPase associated with the membrane, found that RhoGDI can extract both inactive and active RhoGTPases, and found that extraction of active RhoGTPase contributes to their spatial regulation around cell wounds. These results indicate that RhoGDI directly contributes to the spatiotemporal patterning of RhoGTPases by removing active RhoGTPases from the plasma membrane.

**\*For correspondence:**
peter.bieling@mpi-dortmund.
mpg.de (PB);
wmbement@wisc.edu (WMB)

†These authors contributed equally to this work

**Competing interests:** The authors declare that no competing interests exist.

## Introduction

The Rho family GTPases, including Rho, Rac and Cdc42, are essential signaling proteins that mediate morphological changes in cells by directing local cytoskeletal rearrangements (*Bishop and Hall, 2000*; *Kimura et al., 1996*). These rearrangements are generally initiated at and confined to specific subcellular regions. For example, a narrow, concentrated zone of Rho activity directs the formation of a ring of actin filaments and myosin-2 at the equatorial cortex that drives cytokinesis (*Bement et al., 2005*; *Yonemura et al., 2004*; *Yüce et al., 2005*). Similarly, Rho, Rac and Cdc42 are activated near the leading edge of crawling cells in patterns that correspond to local cycles of protrusion, adhesion and retraction (*Machacek et al., 2009*; *Martin et al., 2016*). Because tight spatiotemporal regulation of the GTPases is a fundamental feature of these cellular processes, considerable effort has been invested in studying GTPase regulation.

The RhoGTPases are classically characterized as cycling between membrane-associated, active states and soluble, inactive states as a result of interactions with three classes of regulatory proteins: guanine nucleotide exchange factors (GEFs), which activate GTPases by promoting exchange of GDP for GTP (*Rossman et al., 2005*); GTPase activating proteins (GAPs), which inactivate GTPases by promoting GTP hydrolysis (*Moon and Zheng, 2003*); and guanine nucleotide dissociation inhibitors (GDIs), which solubilize GTPases to generate a large reservoir of heterodimeric GTPase:GDI complexes in the cytoplasm (*Garcia-Mata et al., 2011*). In the canonical model of GTPase regulation, GTPase cycling is thought to proceed as follows: a GTPase is activated by a GEF at the plasma membrane following its release from GDI, is subsequently inactivated by a GAP, and is then returned

**eLife digest** Organisms rely on many signaling molecules to control how their cells grow, divide and heal. For example, when the cell membrane is damaged, two signaling proteins, Rho and Cdc42, are recruited to wounds and activated to promote repair. Active Rho and active Cdc42 form two concentric rings at the membrane to direct the closure of the wound.

Rho and Cdc42 belong to the RhoGTPase family, a group of proteins that act as molecular switches and alternate between active and inactive forms. At the level of the cell, RhoGTPases are only active in the tiny patches of the membrane where they bind. However, individual proteins hop on and off membranes in a matter of seconds, only staying bound for short periods. This mechanism is controlled by a regulatory protein known as RhoGDI, and it allows RhoGTPases to form precise patterns of activity at membranes – such as the rings that surround a wound site. However, it was not known exactly how RhoGDI regulates the activity of RhoGTPases over space and time, partly because it is difficult to study these proteins in the laboratory.

To fill this knowledge gap, Golding, Visco et al. developed new fluorescent probes to track Rho and Cdc42 in wounded cells from frogs and on artificial membranes. The experiments showed that pools of inactive Cdc42 accumulated on membranes, alongside the active form of the protein. RhoGDI removed both active and inactive RhoGTPases from artificial and frog cell membranes. In fact, removing active Rho and Cdc42 proteins from the cell membrane was necessary to form the spatial patterns of RhoGTPase activity observed in wounded frog cells.

The findings of Golding, Visco et al. help to understand how RhoGDI proteins regulate RhoGTPases and provide new tools to further study these proteins. In humans, mutations in either RhoGDI or Cdc42 are responsible for severe conditions such as Nephrotic Syndrome Type 8 or Takenouchi-Kosaki syndrome. In the future, this work may aid the development of treatments and cures for these conditions.

to the soluble pool by GDI. Thus, the lifetime of GTPase activity at the plasma membrane is generally thought to be controlled entirely by GEFs and GAPs, with GDIs essentially serving as passive shuttles that interact exclusively with inactive GTPases.

The notion that RhoGDIs work as passive shuttles rests largely on two findings. First, when GTPase:GDI complexes are purified from cell lysates, the great majority of GTPase within the complex is in the inactive, GDP-bound form (*Abo et al., 1994*), as expected if GDI solubilizes GTPases after inactivation by a GAP. Second, binding of GTPases by GDI strongly suppresses GTP hydrolysis (*Hart et al., 1992*), indicating that hydrolysis must precede extraction from the membrane. However, work by several labs has shown that GDIs bind both inactive and active GTPases with relatively high affinity in vitro (*Hancock and Hall, 1993*; *Hart et al., 1992*; *Nomanbhoy and Cerione, 1996*; *Tnimov et al., 2012*), leading to the suggestion that GDIs may interact with active as well as inactive GTPases in vivo. As such, GDIs might have the potential to exert a more direct role in the regulation of GTPase activity than currently appreciated.

In contrast to GEFs and GAPs, the biochemical activities of GDIs are not well understood. Presently, no consensus exists as to (1) the mechanism by which GDIs extract GTPases from membranes (*Johnson et al., 2009*; *Zhang et al., 2014*), (2) whether GDIs interact with GTPases in a nucleotide-specific manner (*Nomanbhoy and Cerione, 1996*; *Tnimov et al., 2012*), or (3) how GDI activity is coordinated with GEFs or GAPs (*Garcia-Mata et al., 2011*). This uncertainty stems in part from experimental limitations in studying GTPase:GDI dynamics. First, fusion with fluorescent proteins at the amino terminus can impair GTPase localization (*Yonemura et al., 2004*; *Yüce et al., 2005*) and function (*Watson et al., 2014*; *Freisinger et al., 2013*; *Howell et al., 2012*; *Bendezú et al., 2015*; *Lee et al., 2015*; *Coll et al., 2007*) while fusion with the carboxyl-terminus prevents prenylation (*Howell et al., 2012*). In the absence of direct visualization, GTPase dynamics had to be inferred from activity probes. Second, with a few important exceptions (*Johnson et al., 2009*; *Nomanbhoy et al., 1999*), in vitro studies of GTPase:GDI dynamics have utilized unprenylated GTPases, omitted membranes, or both. Additionally, nearly all of these reconstitution experiments focused on the effect of GDI on membrane-associated or soluble GTPases at thermodynamic equilibrium (*Zhang et al., 2014*). Thus, we do not currently understand how GDI affects the transitions

between membrane and soluble GTPase states kinetically. This is especially true under conditions which mimic the cellular environment, which is far from equilibrium due to the constant dissipation of energy.

To overcome these limitations, we developed two distinct methods to directly visualize vertebrate RhoGTPases in vivo and on supported lipid bilayers in vitro. Using these tools, we identify co-existing pools of active and inactive GTPases associated with the plasma membrane and provide additional evidence that GDI can extract both inactive and active GTPases in vitro, as suggested by previous steady-state affinity measurements (*Hancock and Hall, 1993*; *Leonard et al., 1992*; *Nomanbhoy and Cerione, 1996*; *Tnimov et al., 2012*). Finally, we show that the extraction of active GTPase also occurs in vivo and that this contributes to the spatial regulation of GTPase activity. Collectively, these results indicate that GDI itself can indeed directly mediate the spatiotemporal regulation of GTPase activity.

## Results

### Visualization of RhoGTPases around single-cell wounds

Fusion of fluorescent proteins with the carboxy-terminus of RhoGTPases prevents prenylation, while fusion with the amino-terminus can impair their localization and function. This is likely due to occlusion of the switch I/II regions of the GTPase, which is the binding interface for all of its protein:protein interactions (*Dvorsky and Ahmadian, 2004*). Regardless of the explanation, it has been demonstrated that GFP- and YFP-Rho fail to concentrate at the cell equator during cytokinesis in mammalian cells (*Yonemura et al., 2004*; *Yüce et al., 2005*). Further, in gene replacement studies in yeast, amino-terminally tagged Cdc42 fails to localize properly, yields temperature sensitivity and/or aberrant morphology (*Bendezú et al., 2015*; *Coll et al., 2007*; *Freisinger et al., 2013*; *Howell et al., 2012*; *Lee et al., 2015*; *Watson et al., 2014*). We have also found that mCh-Rho, Rac and Cdc42 are not properly recruited to wounds in *Xenopus laevis* oocytes. Specifically, the zones of recruitment defined by amino-terminally tagged GTPases are much less focused and much less intense than those obtained with either the activity reporters or the internally-tagged GTPases (*Figure 1—figure supplement 1*; see below for functional analysis).

To overcome this problem, we first adapted an approach described by *Bendezú et al. (2015)* for labeling of yeast Cdc42. We inserted GFP into a solvent-exposed external loop of the *X. laevis* GTPases (see Methods). To test the internally-tagged (IT) GTPases in vivo, we exploited the cell wound repair model in *X. laevis* oocytes where wounding elicits a robust accumulation of active Rho and Cdc42 in discrete, concentric zones at the cortex as previously indicated by GTPase activity reporters (*Figure 1A*; *Benink and Bement, 2005*). It is important to note that (1) IT-GTPases were co-expressed with wild-type (WT) GDI to avoid disrupting the GTPase:GDI stoichiometric ratio, thereby preventing GTPase aggregation (*Boulter et al., 2010*), and (2) IT-GTPases were expressed at the minimal level necessary to detect signal around the wound (36% above endogenous Rho, based on proteomic data from *Wühr et al., 2014*) to avoid potential overexpression phenotypes (see Materials and methods; *Figure 1—figure supplement 2*). Both IT-Rho and IT-Cdc42 were recruited to concentric rings around the wound (*Figure 1B,C*). Comparison of IT-Rho to a Rho activity reporter (mRFP-2xrGBD; *Davenport et al., 2016*) revealed that IT-Rho spatially overlapped with the Rho activity zone. Comparison of IT-Cdc42 to a Cdc42 activity reporter (mRFP-wGBD; *Benink and Bement, 2005*) revealed that IT-Cdc42 localized throughout the active Cdc42 zone, as well as extended slightly beyond it towards the wound center (see also below). We also tested the behavior of IT-Rac and found that it concentrated around wounds in the same region as IT-Cdc42, as expected from previous experiments (*Figure 1—figure supplement 3*; *Abreu-Blanco et al., 2014*; *Benink and Bement, 2001*).

As an alternative approach, and as a means to obtain labeled RhoGTPases that could be used both in vivo and in vitro, purified recombinant Rho and Cdc42 were prenylated, coupled to Cy3 via a short N-terminal peptide by sortase-mediated ligation, and bound to GDI (see Materials and methods). Cy3-Rho and Cy3-Cdc42, bound to GDI for stabilization, were microinjected into oocytes 41% and 53% above endogenous levels, respectively. Both Cy3-Rho and Cy3-Cdc42 localized to wounds (*Figure 1D,E*), in a manner indistinguishable from their IT counterparts expressed in the oocyte (*Figure 1F,G*). As observed with IT-Rho and IT-Cdc42, Cy3-Rho completely

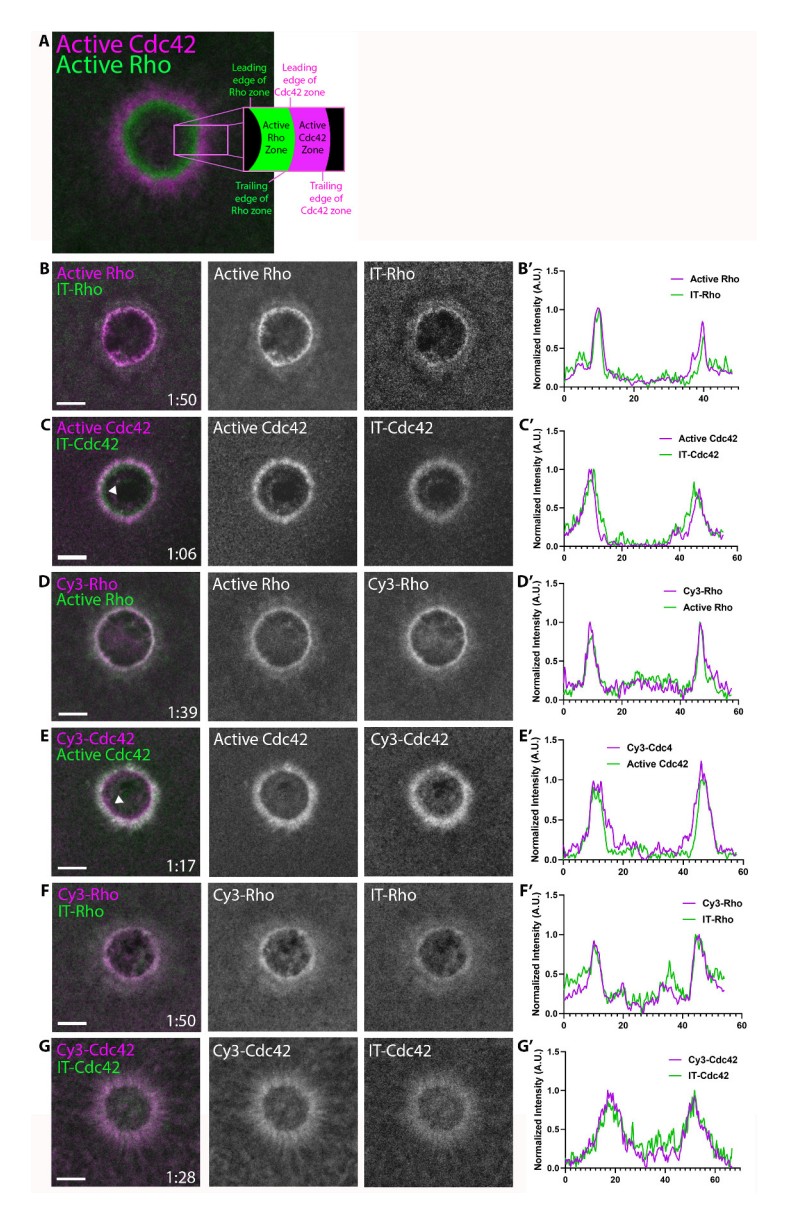

**Figure 1.** Direct visualization of Rho and Cdc42 during cell wound repair. (**A**) Left: image of active Cdc42 (magenta) and active Rho (green) around single-cell wound in *Xenopus laevis* oocyte; right: schematic diagram indicating zone regions; (**B**) Wound in oocyte microinjected with rGBD (active Rho, magenta) and IT-Rho (green); (**B'**) Line scan of normalized fluorescence intensity from (**B**); (**C**) As in B but with wGBD (active Cdc42, magenta) and IT-Cdc42 (green); (**D,D'**) As in B but with Cy3-Rho (magenta) and rGBD (green); (**E,E'**) As in B but with Cy3-Cdc42 (magenta) and wGBD (green); (**F,F'**) As in B but with Cy3-Rho (magenta) and IT-Rho (green); (**G,G'**) As in B but with Cy3-Cdc42 (magenta) and IT-Cdc42 (green) and line scan. Scale bar 10 μm, time min:sec.

The online version of this article includes the following source data and figure supplement(s) for figure 1:

**Source data 1.** Direct visualization of Rho and Cdc42 during cell wound repair.
**Figure supplement 1.** Amino-terminally tagged RhoGTPases do not localize properly to wounds.
**Figure supplement 1—source data 1.** Amino-terminally tagged RhoGTPases do not localize properly to wounds.
**Figure supplement 2.** Expression level of Rho internally-tagged with GFP.
**Figure supplement 2—source data 1.** Expression level of Rho internally-tagged with GFP.
**Figure supplement 3.** Internally-tagged Rac localizes to wounds.
**Figure supplement 3—source data 1.** Internally-tagged Rac localizes to wounds.

overlapped with the zone of Rho activity while Cy3-Cdc42 localized throughout and slightly interior to the active Cdc42 zone. These results indicate that the IT- and Cy3-tagged GTPase variants faithfully mimic endogenous GTPases during cell wound repair.

## Visualization of RhoGTPases in other cellular contexts

To further test the behavior of the IT- and Cy3-labeled RhoGTPases, we sought to determine if they localize to the plasma membrane in other cellular processes. This is important because these processes likely depend on different regulators from those that operate during cell wound repair. IT-Rho and Cy3-Rho localized to the cytokinetic apparatus and epithelial junctions in *Xenopus* embryos, consistent with previous results obtained with a Rho activity reporter (*Figure 2A–C*; *Bement et al., 2005*). Similarly, IT-Cdc42 localized to exocytosing cortical granules (*Figure 2D*), consistent with previous results obtained with a Cdc42 activity reporter (*Yu and Bement, 2007*). IT-Cdc42 was also recruited to cell-cell junctions and enriched there upon wounding (*Figure 2E*), a behavior previously revealed using a Cdc42 activity reporter (*Clark et al., 2009*).

Next, we wanted to determine whether IT-RhoGTPases can functionally substitute for the endogenous GTPases. The *X. laevis* oocyte system is not conducive to traditional knockdown approaches due to its large stores of maternal protein and relatively slow protein turnover. Therefore, we employed C3-exotransferase, a Rho-specific toxin, to inhibit endogenous Rho activity, and expressed an IT-Rho in which the C3 ribosylation site (N41) is mutated to a residue that cannot be ribosylated (N41V; *Sekine et al., 1989*). In control oocytes expressing the probe for active Rho, Rho activity around wounds was suppressed by C3 (*Figure 2F,G*), while cells expressing IT-Rho-N41V generated a spatially defined zone of Rho activity around the wound that closed over similar timescales as the control. While there is a slight decrease in Rho activity in IT-Rho-N41V cells exposed to C3 versus without, we attribute the difference to the loss in endogenous Rho activity. In contrast, cells expressing amino-terminally tagged Rho (mCh-Rho-N41V) failed to rescue Rho activity upon inhibition of endogenous Rho by C3 (*Figure 2H*, *Figure 2—figure supplement 1*). Collectively, these results indicate that, unlike amino-terminally tagged GTPases, both IT- and Cy3-labeled GTPases are faithful reporters of the distribution of GTPases and show that IT-Rho can functionally substitute for its endogenous counterpart.

## Pools of active and inactive RhoGTPase accumulate in the plasma membrane

The observation that Cdc42 extends slightly interior to its zone of activity suggests that there may be a pool of inactive, membrane-bound Cdc42 at this location. This notion is consistent with the previous observation that Abr, a Cdc42-GAP thought to regulate Cdc42 activity at wounds, also localizes interior to the Cdc42 zone (*Vaughan et al., 2011*). To understand the relationship between activity and membrane-association of Cdc42, we overexpressed Abr, a manipulation previously shown to decrease Cdc42 activity around wounds (*Vaughan et al., 2011*). Remarkably, this resulted in a dose-dependent loss of active Cdc42 at wounds while having far less effect on Cy3-Cdc42 (*Figure 3A,B*). These results further demonstrate that the IT- and Cy3-GTPases are functional. More importantly, they demonstrate that substantial pools of both active and inactive GTPases can be dynamically maintained at the plasma membrane.

## RhoGDI is recruited to concentrated areas of RhoGTPase activity

Efforts to visualize RhoGDI at the plasma membrane have generally failed (*Ngo et al., 2017*), likely because GDI only transiently interacts with GTPases upon release into or extraction from the membrane. However, we reasoned it might be possible to detect GDI at wound sites due to the high local concentration of Rho and Cdc42. Indeed, we found that 3xGFP-GDI is enriched at wounds and forms a broad zone that peaks between active Rho and active Cdc42 (*Figure 4A,B*). To confirm that endogenous GDI also localizes to wounds, antibodies were raised against *X. laevis* GDI (*Figure 4—figure supplement 1*) and used to immunolabel wounded oocytes. Consistent with the results obtained with 3XGFP-GDI, endogenous GDI accumulated at wounds (*Figure 4C*). These results demonstrate that GDI accumulation occurs at discrete regions of the plasma membrane that are enriched with its GTPase clients.

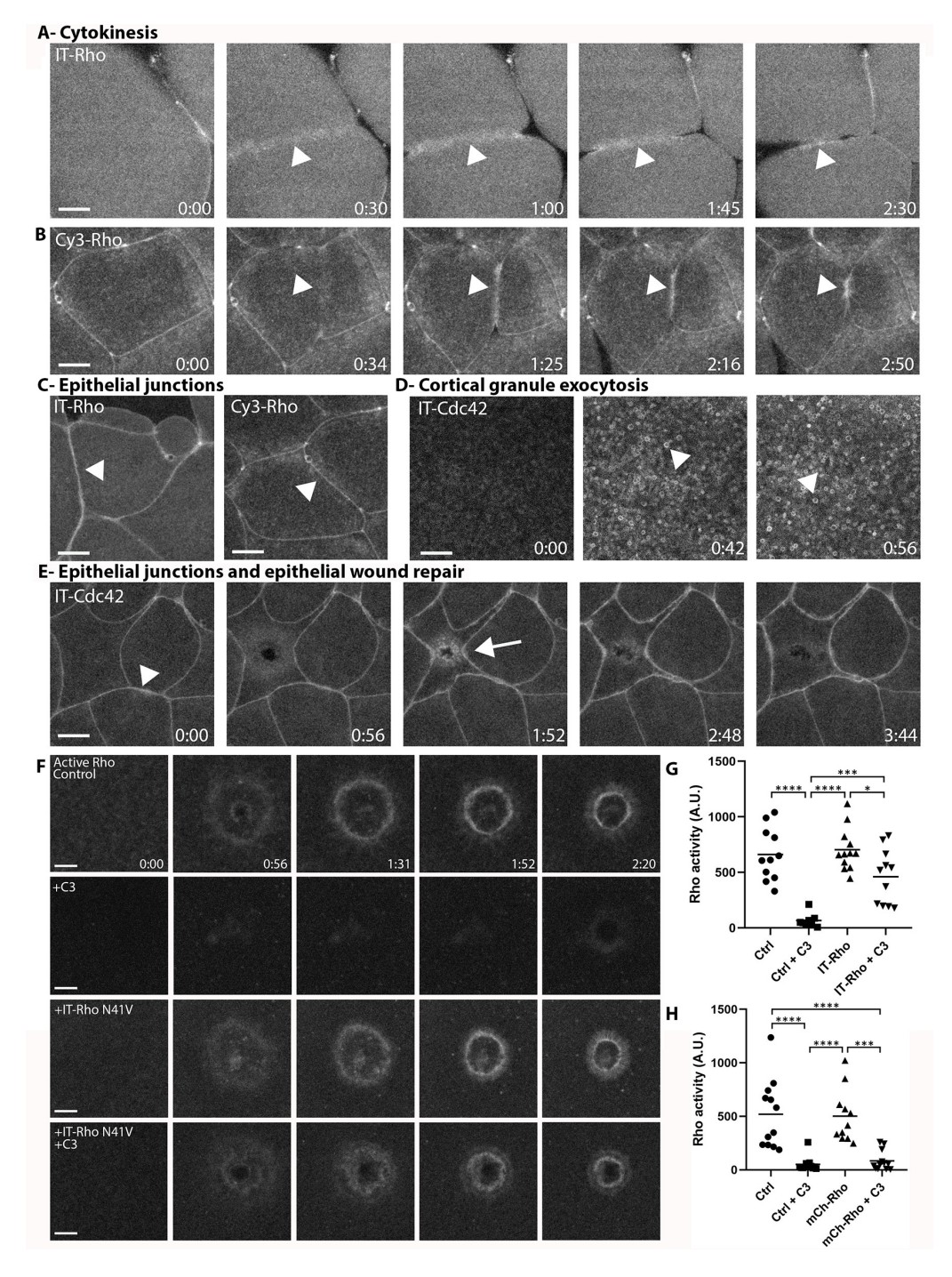

**Figure 2.** Directly-labeled Rho and Cdc42 during cytokinesis, cortical granule exocytosis, epithelial wound repair and at junctions. (A) Cytokinesis in *X. laevis* embryo microinjected with IT-Rho; IT-Rho accumulates at nascent cleavage furrow (arrowhead); (B) Cytokinesis in *X. laevis* embryo microinjected with Cy3-Rho; Cy3-Rho accumulates at nascent cleavage furrow (arrowhead); (C) *X. laevis* embryo microinjected with IT-Rho (left) and Cy3-Rho (right); both are enriched at cell-cell junctions; (D) Meiotically mature *Xenopus* egg microinjected with IT-Cdc42; IT-Cdc42 is recruited to exocytosing cortical granules (arrowheads) following egg activation (0:42); (E) *X. laevis* embryo microinjected with IT-Cdc42; IT-Cdc42 concentrates at cell-cell junctions (arrowhead; 0:00) and, following damage, is recruited to the wound and becomes enriched at junctions (arrow); (F) C3-insensitive IT-Rho rescues Rho activity in presence of C3. Control: oocyte microinjected with rGBD shows normal Rho activation and wound closure; C3: cell microinjected with rGBD fails to activate Rho in presence of C3; IT-Rho-N41V: cell microinjected with rGBD and C3-insensitive IT-Rho activates Rho normally; IT-Rho-N41V+C3: cell microinjected with rGBD and C3-insensitive IT-Rho rescues Rho activity in presence of C3. Scale bar 10 μm, time min:sec; (G) Quantification of Rho

*Figure 2 continued on next page*

*Figure 2 continued*

activity, corrected for background (n = 8–12); (H) As in G but with amino-terminally tagged mCh-Rho (n = 8–9). One-way ANOVA with Tukey post-test statistical analysis. *p<0.05, ***p<0.001, ****p<0.0001.

The online version of this article includes the following source data and figure supplement(s) for figure 2:

**Source data 1.** Directly-labeled Rho and Cdc42 during cytokinesis, cortical granule exocytosis, epithelial wound repair and at junctions.
**Figure supplement 1.** Amino-terminally tagged Rho fails to rescue Rho activity upon inhibition of endogenous Rho.

## RhoGDI differentially regulates rho and Cdc42

The localization of RhoGDI around wounds suggests that it might play an active role in delivery to or extraction of GTPases from the membrane and thus their spatiotemporal patterning. As an initial test of this possibility, we overexpressed GDI via mRNA microinjection. This manipulation potently suppressed both Rho and Cdc42 activity, as well as Cy3-Rho and Cy3-Cdc42 localization, suggesting that GDI exerts its effects via extraction of the GTPases (*Figure 5A*). To obtain a more quantitative understanding of the relationship between GDI and GTPase activity, we microinjected purified GDI (*Figure 5—figure supplement 1*) at increasing concentrations prior to wounding. High concentrations of microinjected GDI suppressed both Rho and Cdc42 activity at wounds (*Figure 5B,C*), consistent with the results obtained from GDI via mRNA-mediated overexpression. However, more modest increases revealed differential effects on Rho and Cdc42. Specifically, slight increases (15.8%) in GDI levels resulted in a greater reduction of Cdc42 activity compared to Rho (*Figure 5C, D*). We found the same to be true for bovine GDI (*Figure 5—figure supplement 2*). These results show that GDI differentially impacts Rho and Cdc42 activity in vivo and that this effect does not require gross overexpression.

## RhoGDI extracts active and inactive RhoGTPase in vitro

To directly probe the mechanism by which RhoGDI inhibits Rho and Cdc42 activity in vivo, we established a real-time GTPase dissociation assay on supported lipid-bilayers (SLBs) (*Figure 6A*, *Figure 6—figure supplement 1*; see Materials and methods). Cy3-Cdc42 was added to SLBs and binding was detected by total internal reflection microscopy (TIRF). Binding was dependent on its C-terminal prenyl moiety, as expected (*Figure 6B*). We then studied the time course of Cdc42 release from SLBs under buffer flow which continuously flushed out unbound proteins from solution. Spontaneous release of inactive, GDP-bound Cdc42 from the membrane was rather slow ($t_{1/2} = 37.24 \pm 3.05$ s), however the addition of excess GDI lead to a dramatic acceleration of dissociation (ca.20-fold; *Figure 6C*, *Figure 6—figure supplement 2*). To determine whether this was the result of either simple sequestration in solution or, alternatively, direct extraction of Cdc42 from membranes, we performed assays in the presence of an alternative solubilizer (RabGGTase beta) that sequesters the GTPase prenyl moiety (*Figure 6C*). Sequestration alone only marginally affected the rate of dissociation, demonstrating that GDI directly extracts GTPases from membranes (*Figure 6D,E*).

To characterize membrane extraction more quantitatively, we carried out experiments over a wide range of RhoGDI concentrations using either inactive (GDP-bound) or active (GTP-bound, constitutively-active) forms of Cdc42 or Rho. We observed that WT GTPases hydrolyze even GTP analogs such as GTPγS over the long time period (hours) required for performing a full titration in our SLB assay (data not shown), which led to them accumulating in the inactive, GDP-bound form during the course of the experiment. We therefore turned to two constitutively-active GTPase variants: Q61L and G12V for Cdc42 (Q63L and G14V for Rho). While often used interchangeably, the biochemical properties of these two mutants are not entirely equivalent (*Smith et al., 2013*). The Q61L substitution directly and strongly impairs spontaneous nucleotide hydrolysis, whereas the G12V mutation only moderately inhibits spontaneous GTPase activity. The constitutive activity of G12V variants rather originates from defects in GAP-induced stimulation of GTP hydrolysis. Previous work suggests that RhoGDI interacts much more weakly with Q61L variants compared to G12V in cells (*Hodgson et al., 2016*; *Michaelson et al., 2001*; *Pertz et al., 2006*). Whether this is due to these mutations differentially weakening affinity for GDI directly or, alternatively, indirect effects resulting from differences in the nucleotide states these variants assume in the cytoplasm is unknown. We therefore tested the extraction of both of these variants in an active state from membranes by GDI.

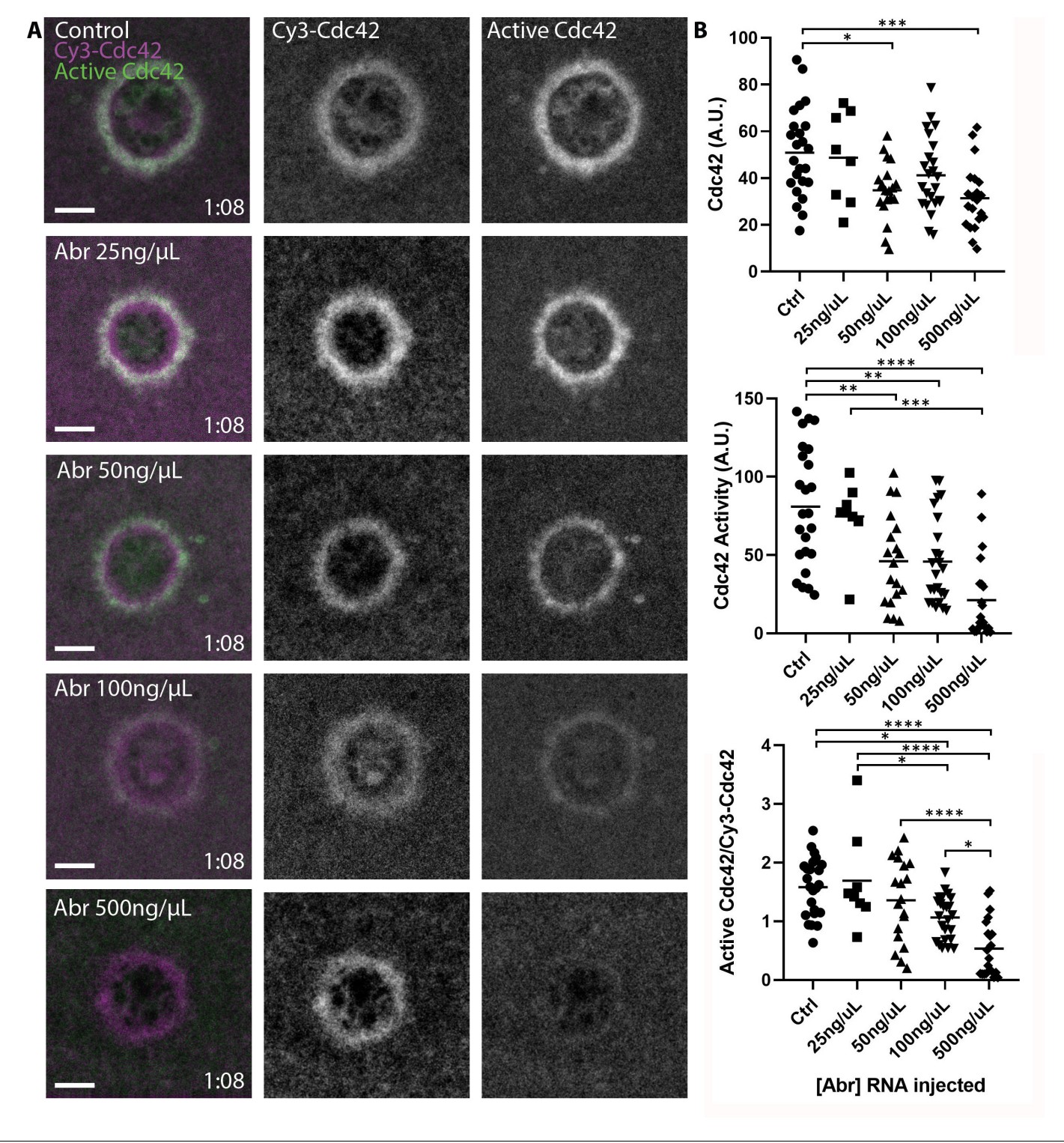

**Figure 3.** Pools of inactive and active Cdc42 at the plasma membrane. (**A**) Oocytes microinjected with wGBD (green), Cy3-Cdc42 (magenta) and indicated concentrations of mRNA encoding the Cdc42-GAP Abr. Scale bar 10 μm, time min:sec.; (**B**) Quantification of Cy3-Cdc42, active Cdc42 and ratio of active Cdc42 to Cy3-Cdc42 for each condition. n = 8–24. One-way ANOVA with Tukey post-test statistical analysis. *p<0.05, **p<0.01, ***p<0.001, ***p<0.0001.

The online version of this article includes the following source data for figure 3:

**Source data 1.** Pools of inactive and active Cdc42 at the plasma membrane.

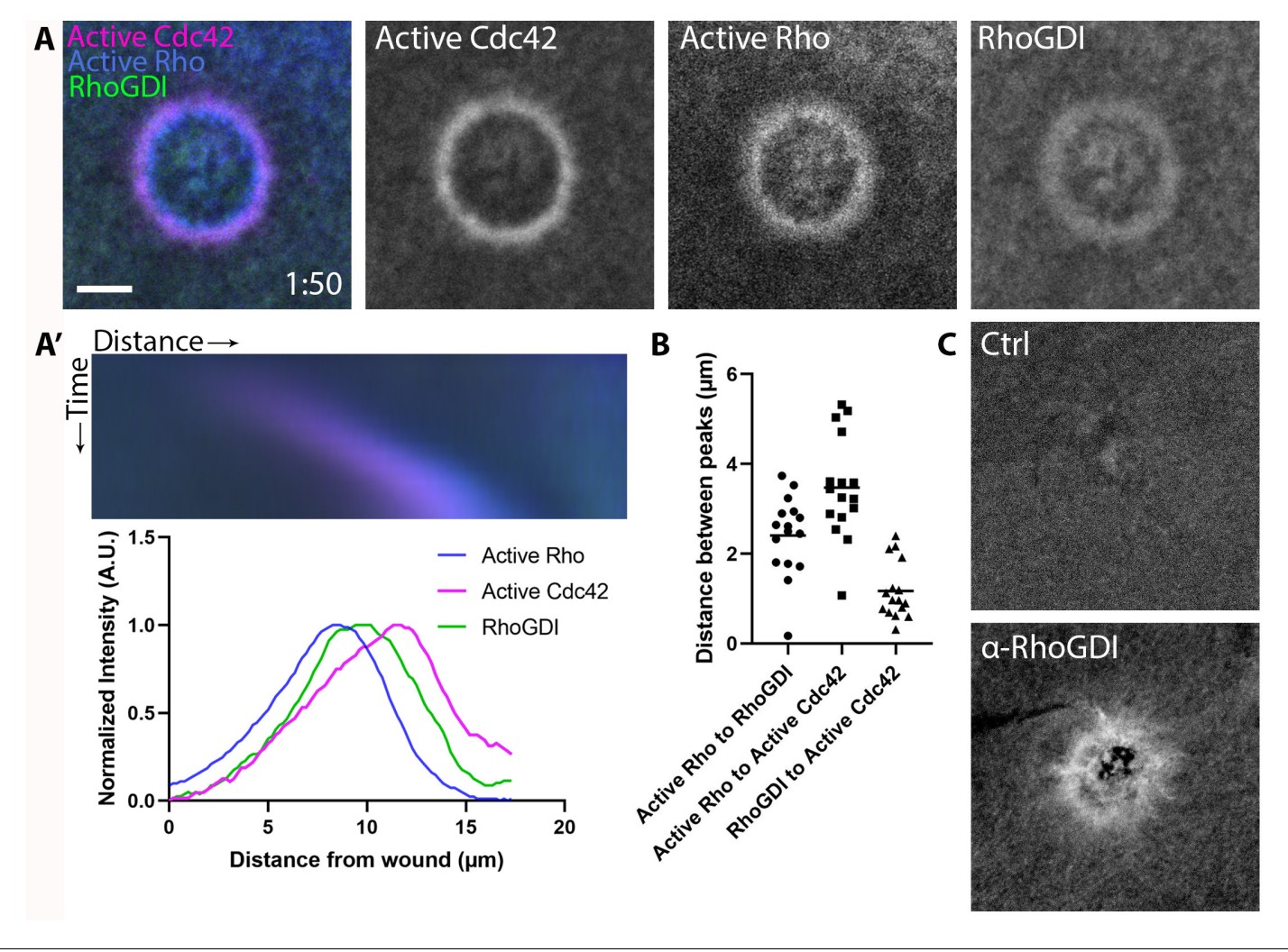

**Figure 4.** RhoGDI is recruited to single-cell wounds enriched in Rho and Cdc42 activity. (**A**) Oocytes microinjected with wGBD (magenta), rGBD (blue) and GDI (green); (**A'**) Kymograph of wound closure and line scan of radially-averaged fluorescence intensity from (**A**); (**B**) Quantification of distance between peaks (n = 16); (**C**) Wounded oocytes fixed and stained with anti-*X. laevis* GDI. Scale bar 10 µm, time min:sec.

The online version of this article includes the following source data and figure supplement(s) for figure 4:

**Source data 1.** RhoGDI is recruited to single-cell wounds enriched in Rho and Cdc42 activity.

**Figure supplement 1.** *X. laevis* RhoGDI antibody specificity.

In line with the known catalytic differences of these RhoGTPase variants (*Smith et al., 2013*), we found that they, when purified recombinantly, assume different nucleotide states as determined by HPLC analysis: Q61L GTPases were exclusively GTP-bound, whereas G12V mutants purified as a mix of GTP and GDP bound states (*Figure 7—figure supplement 1*). To mimic a homogenously active state of G12V variants in vitro, we exchanged their bound GTP and GDP nucleotides for GTPγS directly before the extraction experiment.

Remarkably, RhoGDI was able to extract both inactive (GDP-bound) and active (GTPγS-bound, G12V/G14V) Cdc42 and Rho in a concentration-dependent manner (*Figure 7A,B*). Although GDI extracted GDP-bound GTPase more efficiently than GTPγS-bound, G12V/G14V GTPases (1.5 fold difference), it was still able to effectively facilitate the dissociation of the latter (*Figure 7C,F*). We made qualitatively similar observations for the GTP-bound Q61L/Q63L variants, which were also extracted by GDI, albeit with reduced rates compared to their G12V/G14V counterparts (*Figure 7—figure supplements 2* and *3*). The affinities of GDI for the active and inactive GTPases on membranes, determined by hyperbolic fits to the extraction rates, were surprisingly similar (*Figure 7G,H*,

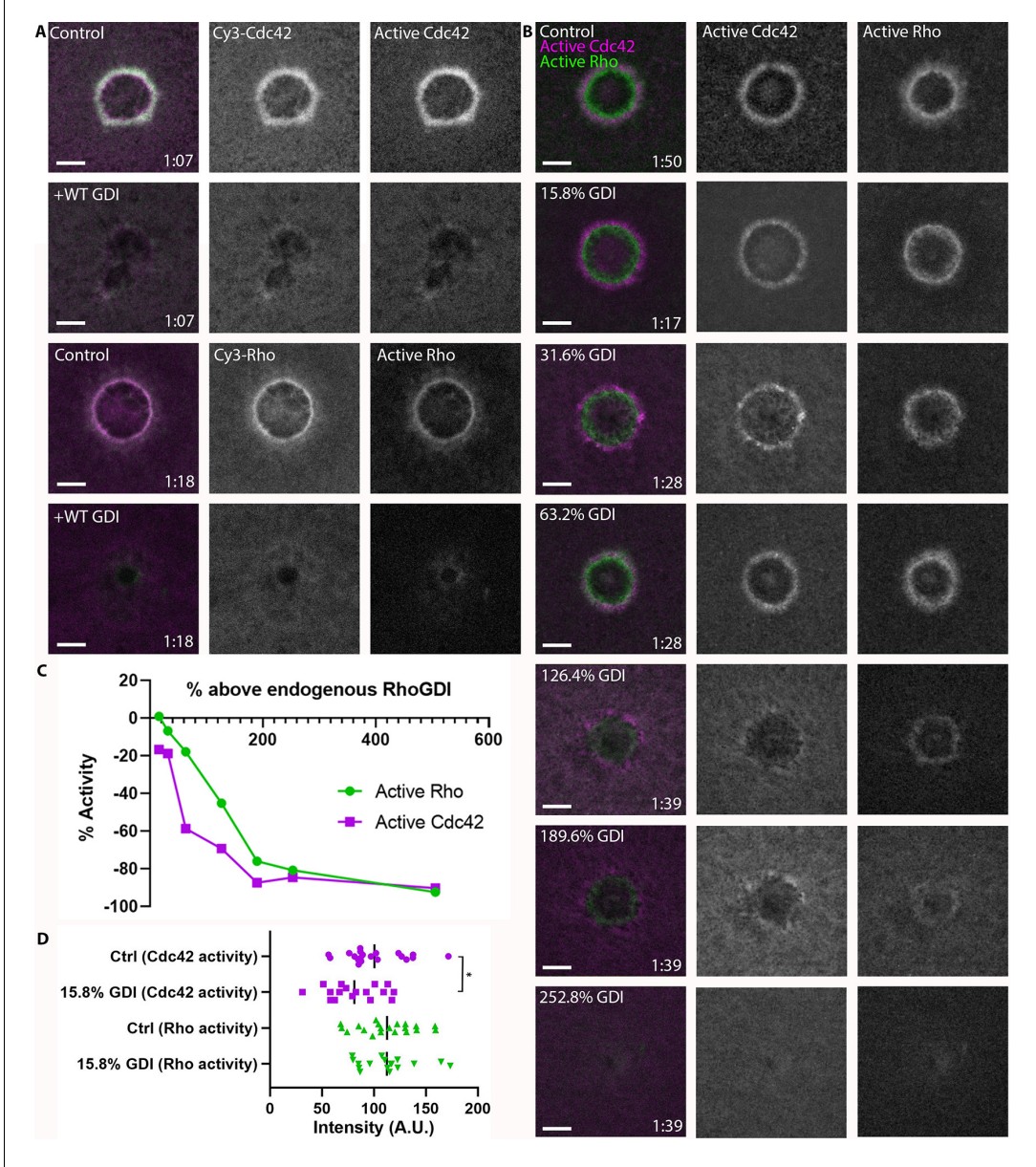

**Figure 5.** RhoGDI overexpression differentially regulates Rho and Cdc42 activity. (**A**) Top two rows: oocytes microinjected with Cy3-Cdc42 (magenta) and wGBD (green) alone or with GDI. Bottom two rows: Oocytes microinjected with Cy3-Rho (magenta) and rGBD (green) alone or with GDI; (**B**) Oocytes microinjected with wGBD (magenta), rGBD (green) and increasing concentrations of GDI protein. Scale bar 10 μm, time min:sec; (**C**) Standard curve of Rho and Cdc42 activity with increasing concentrations of GDI (n = 10–23 for each concentration); (**D**) Quantification of Rho and Cdc42 activity at 15.8% above endogenous GDI (n = 17–20). Unpaired student's t-test, 2-tailed distribution, equal variance statistical analysis. *p<0.05.

The online version of this article includes the following source data and figure supplement(s) for figure 5:

**Figure supplement 1.** Purified *X. laevis* RhoGDI protein.

**Figure supplement 2.** Bovine RhoGDI decreases Rho and Cdc42 activity in a dose-dependent manner in vivo.

**Figure supplement 2—source data 1.** Bovine RhoGDI decreases Rho and Cdc42 activity in a dose-dependent mannerin vivo.

*Supplementary file 1*). On the other hand, the maximal rates of extraction were not, indicating that the rate-limiting step of membrane extraction depends on the activity state of GTPases. To investigate whether extraction of active and inactive GTPases is a conserved ability among GDI proteins, we also studied mammalian GDI. Similar to its *Xenopus* ortholog, bovine GDI1 was able to extract both GDP- as well as GTP-bound Cdc42Q61L and RhoQ63L variants (*Figure 7—figure supplement 4*). These

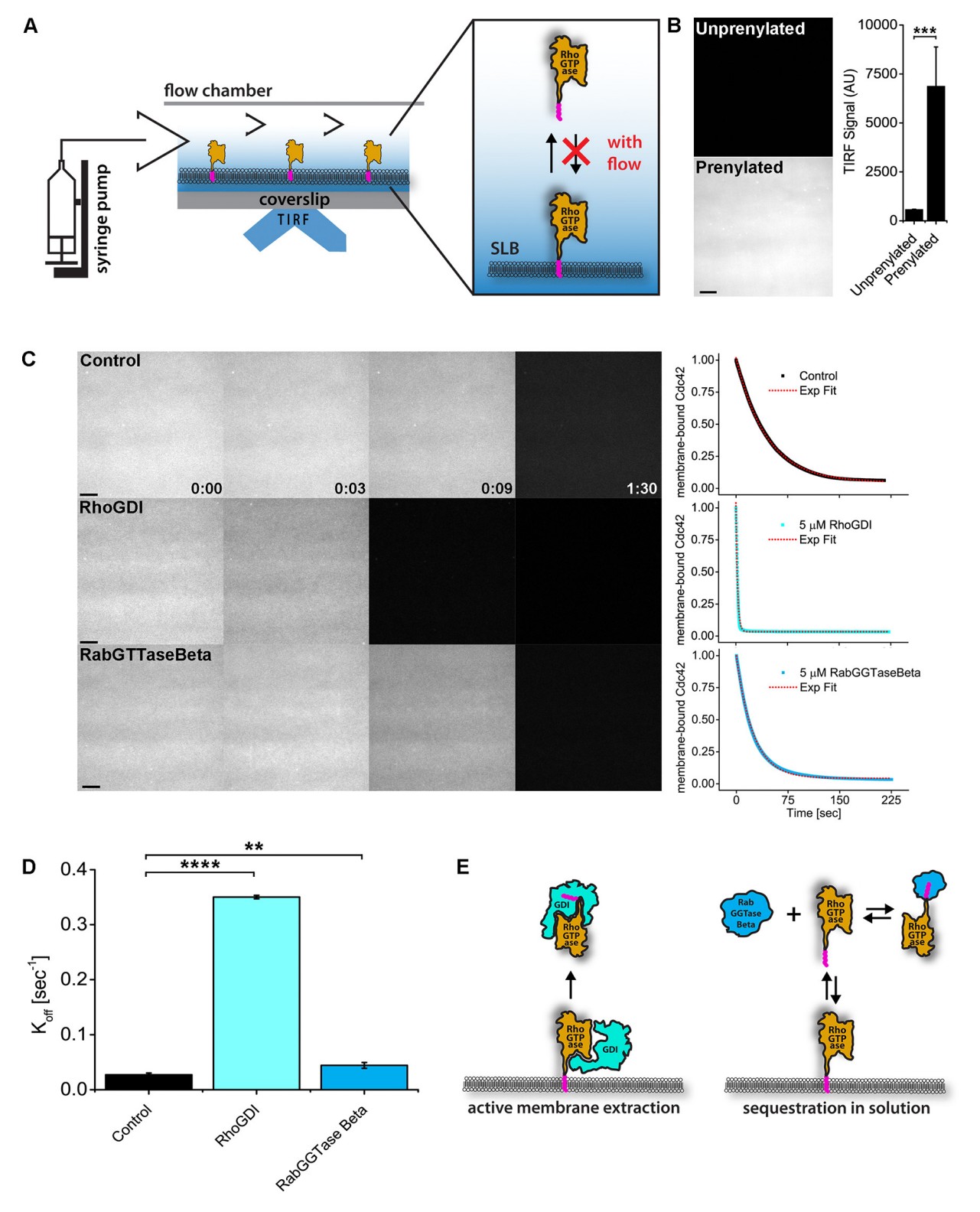

**Figure 6.** RhoGDI extracts RhoGTPases from membranes in vitro. (**A**) Experimental setup of in vitro experiments: prenylated RhoGTPases were reconstituted on supported lipid bilayers (SLBs) in flow chambers and imaged by TIRF. Wash off experiments were designed to avoid RhoGTPase rebinding to membranes and performed controlling the flow rate via a syringe pump; (**B**) TIRF imaging allows for selective imaging of RhoGTPases at the membrane. Prenylated and unprenylated Cdc42 were imaged in the same conditions and TIRF signal at membranes was quantified (n = 4 for each

*Figure 6 continued on next page*

Figure 6 continued

condition); (C) Wash off experiments: prenylated Cdc42 reconstituted on SLBs were washed with imaging buffer only (control), in presence of 5 μM RhoGDI or RabGTTase Beta. Time lapse images at selected time points and quantification of the full experiments are shown. Decay curves were fitted with a monoexponential function; (D) Comparison of the $K_{off}$ values obtained by fitting the decay curves (n = 3 for control and RabGTTase Beta, n = 2 for RhoGDI); (E) Schematic representation of the proposed mode of action of the two RhoGTPases solubilizers. RabGGTase Beta sequesters RhoGTPases in solution, whereas RhoGDI actively extracts RhoGTPases from the membranes. Scale bar 10 μm. Unpaired student's t-test, 2-tailed distribution, equal variance statistical analysis. **p<0.01, ***p<0.001, ****p<0.0001.

The online version of this article includes the following source data and figure supplement(s) for figure 6:

**Source data 1.** RhoGDI extracts RhoGTPases from membranes in vitro.
**Figure supplement 1.** In vitro data analysis.
**Figure supplement 2.** Purified RhoGDI protein.

---

data clearly demonstrate that GDIs can directly extract both inactive and active GTPase from membranes in vitro.

## Identification of an extraction-deficient RhoGDI

The canonical RhoGTPase cycle assumes that GDI does not extract GTPase without its prior inactivation by a GAP (*Garcia-Mata et al., 2011*). However, the above results suggest that GDI might directly attenuate GTPase activity at the plasma membrane via extraction of active GTPase. If this hypothesis is correct, then expression of an extraction-deficient GDI would influence the spatiotemporal patterning of GTPase activity. We therefore sought to generate an extraction-deficient GDI that would still initiate contact with GTPases but fail to extract them from the plasma membrane. Mutants were screened by quantifying their recruitment to wounds relative to WT GDI, based on the rationale that mutants capable of binding but not extracting should remain at the membrane longer and thus accumulate at wounds more than WT GDI.

Using this screen, we first tested RhoGDI mutants that were previously reported to be deficient in extraction (*Dransart et al., 2005*; *Ueyama et al., 2013*). None of these extraction-deficient mutants were recruited to wounds more strongly than WT GDI, suggesting that they were impaired in binding to GTPases at wounds rather than extracting GTPases from wounds (*Figure 8—figure supplement 1*). We therefore designed three novel mutants: (1) the isolated regulatory arm of GDI that initiates contact with the GTPase but lacks a binding pocket for the hydrophobic prenyl group (Δ51–199) and (2,3) mutation of residues E158/9 previously hypothesized to be responsible for GTPase extraction (*Hoffman et al., 2000*). GDI mutants Δ51–199, E158/9A and E158/9Q were halo-tagged, expressed in oocytes and their recruitment to wounds was quantified relative to WT GDI. GDI Δ51–199 showed minimal recruitment to wounds, however both GDI E158/9A and E158/9Q showed a significant increase in recruitment to wounds relative to WT GDI, with GDI E158/9Q (GDI-QQ) having the greatest increase (*Figure 8A,B*).

To directly test whether RhoGDI E158/9Q (QQ) is deficient in extraction, its functional capabilities were tested in vitro in the SLB assay. WT GDI was able to extract both inactive (GDP-bound) Cdc42 as well as active (GTPγS-bound G12V or GTP-bound Q61L) Cdc42 from the supported lipid bilayers (*Figure 7*, *Figure 7—figure supplements 2* and *3*). In contrast, GDI-QQ retained most of its ability (less than two-fold reduction) to extract inactive (GDP-bound) Cdc42 but was completely deficient in extracting active (both GTPγS-bound G12V and GTP-bound Q61L) Cdc42 (*Figure 8C*, *Figure 8—figure supplement 2*, *Figure 7—figure supplement 3C,D*). Similar observations were made for Rho, however GDI-QQ retained some ability to extract active GTPγS-bound G14V, but not GTP-bound Q63L Rho (*Figure 8C*, *Figure 8—figure supplement 2*, *Figure 7—figure supplement 3*). The corresponding mutant of bovine GDI1 (E163/4Q) shows an equivalent behavior to its *Xenopus* ortholog (*Figure 8—figure supplement 3*). These results confirm that the GDI-QQ mutant is indeed extraction deficient: modestly deficient for inactive GTPases, moderately deficient for active Rho and completely deficient for active Cdc42.

## RhoGDI extracts active RhoGTPase in vivo

We sought to directly test whether RhoGDI can extract active GTPases in vivo by employing GDI-QQ. First, we compared the effects of WT vs. QQ GDI overexpression on wounded oocytes expressing constitutively-active Cdc42 (G12V) (Q61L could not be used as it failed to elevate Cdc42 levels

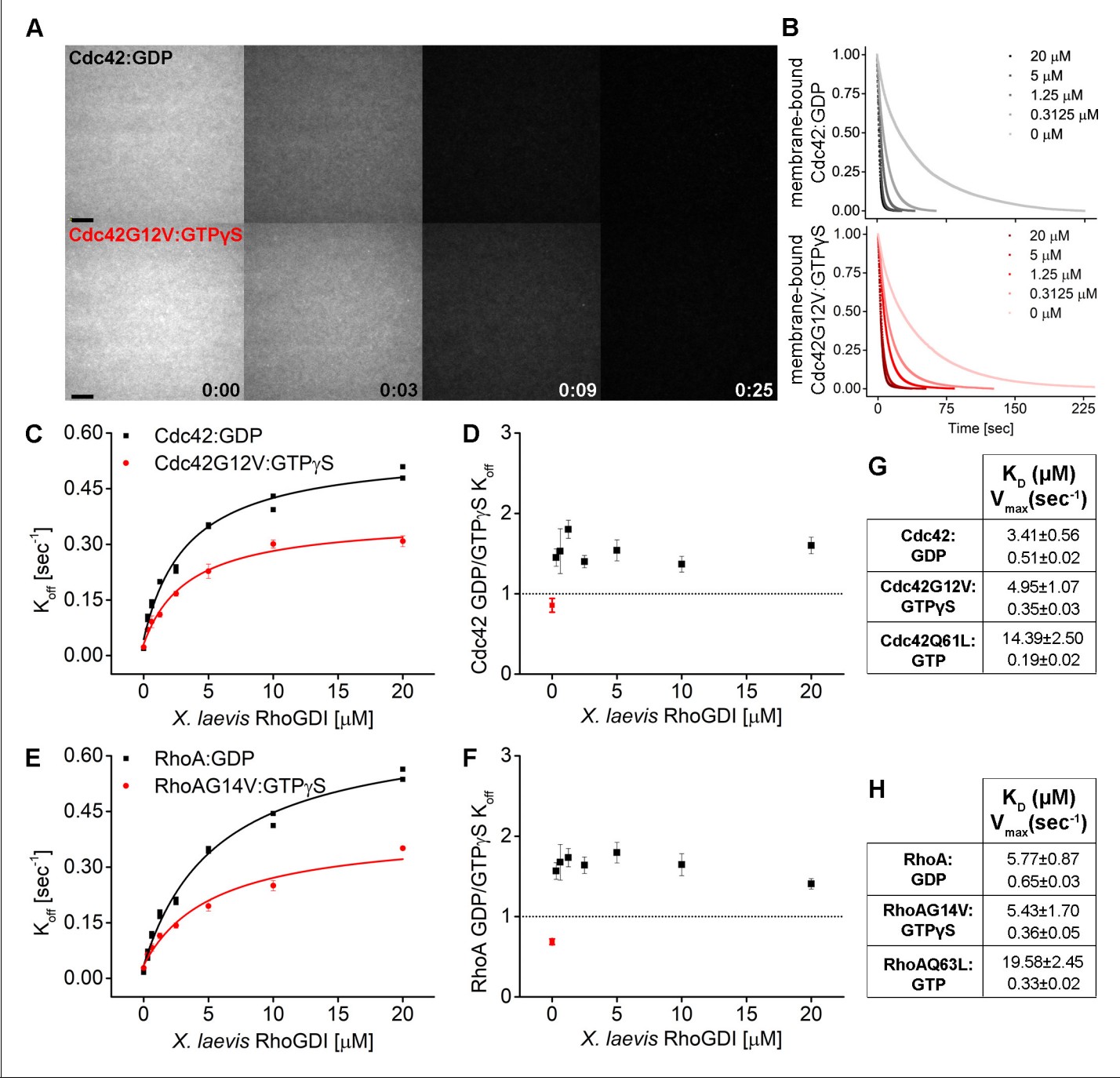

**Figure 7.** RhoGDI extracts both inactive and active RhoGTPases from membranes in vitro. (A) Wash off experiments: prenylated Cdc42 in both inactive (Cdc42:GDP) and constitutively-active (Cdc42G12V:GTPγS) states were reconstituted on SLBs and washed in presence of 5 μM GDI. Time lapse images at selected time points are shown; (B) Quantification of wash off experiments in which the concentration of GDI was titrated between 0 and 20 μM; (C) $K_{off}$ values obtained for inactive and constitutively-active Cdc42G12V fitting the decay curves with a monoexponential decay function are plotted against GDI concentration. Extraction rates were fitted with a hyperbolic function; fitting parameters $K_d$ and $V_{max}$ are reported in table G; (D) Ratio of $K_{off}$ obtained for inactive and constitutively-active Cdc42G12V at the same GDI concentration; (E–F, H) Same as in C-D and G for inactive (Rho:GDP) and constitutively-active (RhoG14V:GTPγS) Rho. Scale bar 10 μm.

The online version of this article includes the following source data and figure supplement(s) for figure 7:

**Source data 1.** RhoGDI extracts both inactive and active RhoGTPases from membranes in vitro.

**Figure supplement 1.** Nucleotide state of constitutively-active RhoGTPase variants after purification.

**Figure supplement 2.** RhoGDI extracts both inactive and active RhoGTPases from membranes in vitro.

*Figure 7 continued on next page*

*Figure 7 continued*

**Figure supplement 2—source data 1.** RhoGDI extracts both inactive and active RhoGTPases from membranes in vitro.
**Figure supplement 3.** Comparison of G12V and Q61L constitutively-active RhoGTPases.
**Figure supplement 4.** Comparison of bovine and *Xenopus* RhoGDI in their ability to extract both inactive and active RhoGTPases from synthetic membranes.
**Figure supplement 4—source data 1.** Comparison of bovine and *Xenopus* RhoGDI in their ability to extract both inactive and active RhoGTPases from synthetic membranes.

around wounds; see *Figure 9—figure supplement 1*). While WT GDI significantly reduced the amount of Cdc42 (G12V) around wounds, GDI-QQ did not (*Figure 9A,B*). Second, we compared the effects of WT vs. QQ GDI overexpression on wounded oocytes microinjected with Cy3-Cdc42 bound to GTPɣS. WT GDI significantly reduced the amount of Cy3-Cdc42(GTPɣS) around wounds while GDI-QQ did not (*Figure 9C,D*). Collectively, these data suggest that GDI can extract active GTPase from the plasma membrane in vivo.

The above results imply that the extraction of active RhoGTPase by GDI might contribute to its spatiotemporal patterning in vivo. To test this hypothesis, we expressed GDI-QQ and monitored the consequences on Rho and Cdc42 activity following wounding. Strikingly, Cdc42 activity around wounds was significantly elevated, in contrast to Rho which was unaffected (*Figure 9E,F*). To assess whether the increase in activity was due to an increase of total Cdc42 around wounds as opposed to competition between GDI-QQ and the Cdc42 activity probe, we repeated the experiment with Cy3-Cdc42. Similar to the results obtained with the activity reporter, expression of GDI-QQ elevated Cy3-Cdc42 levels around wounds relative to controls (*Figure 9G,H*). Cumulatively, these data suggest that GDI directly extracts active Cdc42 throughout the Cdc42 zone, and that extraction of active Cdc42 is necessary for its regulation around wounds.

## Discussion

Direct visualization of the RhoGTPases in living cells is essential for the understanding of their complex spatiotemporal dynamics. We have established two methods to fluorescently label vertebrate GTPases that localize properly: internal tagging with a fluorescent protein and sortase-mediated labeling with a fluorescent dye. This now provides us with widely applicable reagents to analyze GTPase function. These probes faithfully mimic the distribution of the endogenous GTPases based on their comparison to activity reporters in several processes: cell wound repair, cytokinesis, junctional integrity and epithelial wound repair. Further, the successful rescue of Rho function at wounds in the presence of C3 by a C3-insensitive mutant of IT-Rho indicates that these proteins are capable of replacing their endogenous counterparts. It will be important to assess the ability of IT- or Cy3-labeled GTPases to substitute for their endogenous counterparts in other cellular processes in the future via gene replacement approaches, although we note that IT-Cdc42 has been shown to be functional in fission yeast (*Bendezú et al., 2015*). In any case, the combination of the two labeling approaches is powerful as it permits side-by-side comparison of results obtained in vivo and in vitro, as demonstrated here.

Visualization of labeled RhoGTPases in combination with activity reporters in living cells led to an unexpected observation: pools of inactive Cdc42 at the plasma membrane. To the best of our knowledge, this is the first time that inactive GTPases have been detected on membranes under conditions other than gross GTPase overexpression. Notably, the pool of inactive Cdc42 spatially coincides with a local Cdc42-GAP, Abr (*Vaughan et al., 2011*). This pool of inactive Cdc42 expands with overexpression of Abr, further demonstrating that locally-inactivated Cdc42 can remain associated with the plasma membrane. This finding has important mechanistic implications for the regulation of GTPase activity. Namely, it suggests that GTP hydrolysis and extraction of GTPases, while they are likely linked, are not necessarily tightly coupled. This raises the possibility that GTPases might cycle through multiple rounds of activation and inactivation by GEFs and GAPs while remaining associated with the membrane.

Remarkably, in addition to the RhoGTPases themselves, GDI also localized to the plasma membrane in proximity to wounds. The localization of GDI in the same place where the GTPases are especially abundant implies that its accumulation reflects interaction with is GTPase clients. It will be

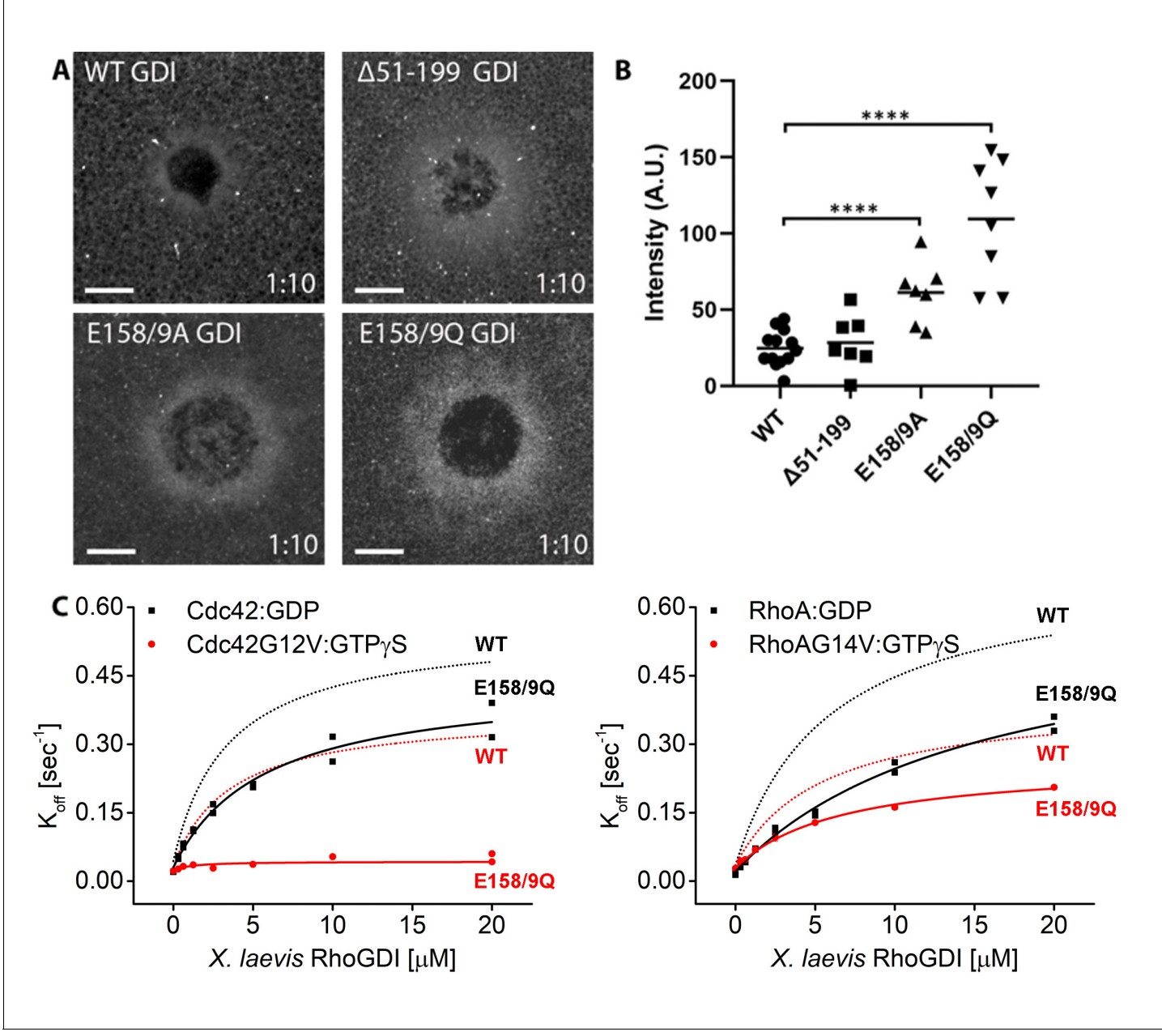

**Figure 8.** Identification of mutant RhoGDI deficient in extraction of active RhoGTPase. (**A**) Oocytes microinjected with halo-tagged WT, Δ51–199, E158/9A and E158/9Q GDI mutants. Scale bar 10 µm, time min:sec; (**B**) Quantification of GDI intensity at wounds (n = 7–13). Unpaired student's t-test, 2-tailed distribution, unequal variance statistical analysis to WT. ****p<0.0001. (**C**) Comparison of $K_{off}$ values obtained for inactive (Cdc42:GDP, Rho:GDP) and constitutively-active (Cdc42G12V: GTPγS, RhoG14V: GTPγS) Cdc42 and Rho from wash off experiments in presence of either WT (black) or E158/9Q (QQ) (red) GDI. Extraction rates were fitted with a hyperbolic function.

The online version of this article includes the following source data and figure supplement(s) for figure 8:

**Source data 1.** Identification of mutant RhoGDI deficient in extraction of active RhoGTPase.

**Figure supplement 1.** Analysis of previously-described extraction-deficient RhoGDI mutants.

**Figure supplement 1—source data 1.** Analysis of previously-described extraction-deficient RhoGDI mutants.

**Figure supplement 2.** Mutant E158/9Q RhoGDI is deficient in extraction of constitutively-active Cdc42Q61L and RhoQ63L in vitro..

**Figure supplement 2—source data 1.** Mutant E158/9Q RhoGDI is deficient in extraction of constitutively-active Cdc42Q61L and RhoQ63Lin vitro.

**Figure supplement 3.** Mutant E163/4Q bovine RhoGDI is deficient in extraction of active Cdc42 and Rho in vitro.

**Figure supplement 3—source data 1.** Mutant E163/4Q bovine RhoGDI is deficient in extraction of active Cdc42 and Rhoin vitro.

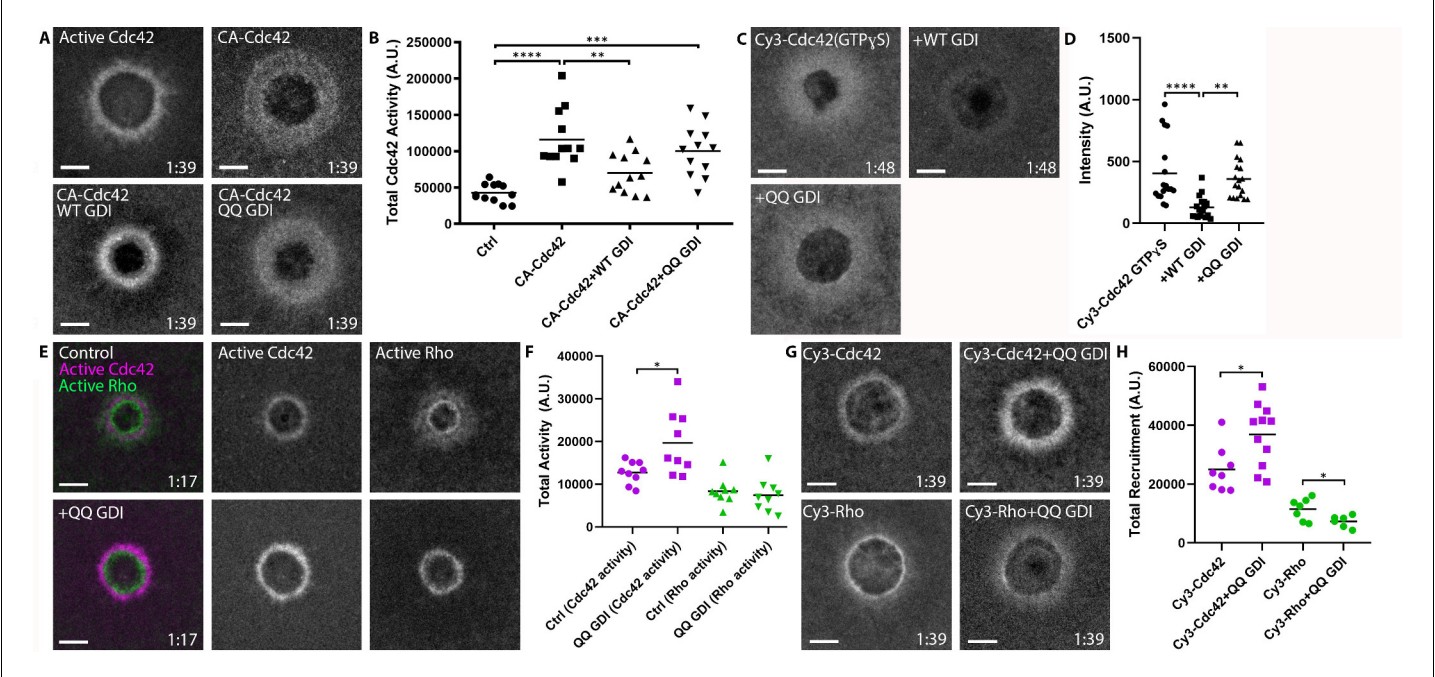

**Figure 9.** RhoGDI extracts active Cdc42 in vivo. (A) Oocytes microinjected with wGBD alone or with constitutively-active Cdc42 (CA:G12V), WT or QQ GDI; B) Quantification of total Cdc42 activity for (A), (n = 12); C) oocytes microinjected with Cy3-Cdc42 bound to GTPɣS alone or with WT or QQ GDI; D) Quantification of intensity for (C), (n = 18). Scale bar 10 μm, time min:sec. One-way ANOVA with Tukey post-test statistical analysis; E) Oocytes microinjected with wGBD (magenta), rGBD (green) alone or with QQ GDI; F) Quantification of total Cdc42 (magenta) and Rho (green) activity from (E) (n = 9); G) Cy3-Cdc42 or Cy3-Rho alone or with QQ GDI; H) Quantification of total recruitment of Cy3-Cdc42 (magenta) and Cy3-Rho (green) (n = 6–11). Scale bar 10 μm, time min:sec. Unpaired student's t-test, 2-tailed distribution, unequal variance statistical analysis. *p<0.05, **p<0.01, ***p<0.001, ***p<0.0001.

The online version of this article includes the following source data and figure supplement(s) for figure 9:

**Source data 1.** RhoGDI extracts active Cdc42 in vivo.
**Figure supplement 1.** Cdc42 Q61L does not behave like constitutively-active Cdc42 in vivo.

important to investigate the detailed mechanism of GDI localization and the control of its turnover at sites of high GTPase activity in the future.

The most significant result of this study is that RhoGDI can extract active GTPase in vivo, particularly active Cdc42 during cell wound repair. This finding is based on two complementary lines of evidence. First, in vitro assays show that GDI directly extracts active GTPases from supported lipid bilayers. This ability is shared between GDI orthologs from distinct species and can be robustly observed for different GTPase mutants trapped in an active configuration (G12/14V and Q61/63L). Their moderate differences in extraction kinetics might nonetheless be biologically meaningful. Previous studies suggest that Q61/63L variants interact more weakly with GDI compared to their G12/14V counterparts (*Hodgson et al., 2016*; *Michaelson et al., 2001*; *Pertz et al., 2006*); we observe a similar trend concerning membrane extraction (*Figure 7—figure supplement 3*).

Second, we also demonstrate extraction of active GTPase in vivo. WT, but not GDI-QQ, extracts Cdc42(G12V) and GTPɣS-Cdc42 from the plasma membrane. These results support previous findings that GDI binds both inactive and active GTPase with relatively high affinity in vitro (*Hancock and Hall, 1993*; *Hart et al., 1992*; *Nomanbhoy and Cerione, 1996*; *Tnimov et al., 2012*). Although we report greater extraction of inactive versus active GTPase, we demonstrate that the extraction of active GTPase is important for their spatiotemporal patterning during cell wound repair. We thus conclude that GDI has the capacity to extract active GTPases and that this ability is harnessed to limit the level of Cdc42 activity during cell wound repair.

This finding has the virtue of explaining previous results in the oocyte cell wound repair system. Based on an indirect approach involving photoactivatable Rho and Cdc42 activity reporters, it was found that Cdc42 activity is lost throughout its zone, while Rho activity is preferentially lost at the

trailing edge of its zone (*Burkel et al., 2012*). The results presented here suggest that GDI is responsible for the removal of active Cdc42 throughout the Cdc42 zone, while Rho is inactivated by a trailing edge GAP prior to extraction (*Figure 10—figure supplement 1*). This may also explain why a mild overexpression of GDI significantly reduced Cdc42 activity but had no effect on Rho activity: loss of active Cdc42 can be controlled at the level of GDI while Rho inactivation is controlled at the level of a GAP. However, there is an alternative explanation for why the expression of GDI-QQ causes an increase in Cdc42 activity but not Rho activity: while GDI-QQ is utterly deficient in extraction of active Cdc42, it retains a modest ability to extract active Rho (*Figure 7*).

The broader implications of RhoGDI's ability to extract active GTPase are two-fold. First, it suggests that a new branch should be added to the canonical GTPase cycle in which active GTPase can directly extracted from the plasma membrane by GDI (*Figure 10*). Further, because GDI binding strongly inhibits GTP hydrolysis and nucleotide exchange (*Hart et al., 1992*; *Ueda et al., 2001*), active GTPase may exist in its soluble form in complex with GDI. However, complementary evidence from biochemical and biological studies suggest that active GTPases are less stably bound to GDI compared to their inactive form (*Hodgson et al., 2016*; *Slaughter et al., 2009*; *Tnimov et al., 2012*). As such, this secondary extraction branch may actually represent a loop through which active GTPases are not only removed from cell membranes, but rapidly returned to them (*Figure 10*). Such a mechanism might enhance the spatial reach of GTPase activity within the plasma membrane or even mediate its spreading between different membrane compartments (*Palamidessi et al., 2008*).

Second, RhoGDI's ability to extract active GTPase forces us to reassess its role in GTPase regulation in different cellular processes. While the field primarily studies local GTPase regulation at the level of GEFs and GAPs, we should reconsider GDI's role in regulation, as well as the regulation of GDI itself. Consistent with this idea, several studies have reported that the differential phosphorylation of GDI promotes global increases in the activity of Rho, Rac or Cdc42 by modulating the affinity of GDI for different GTPases (reviewed by *Garcia-Mata et al., 2011*). The results presented here suggest that it will be of considerable interest to assess the potential for phosphorylation and GDI-dependent regulation on a local level. Indeed, results from empirical (*Vaughan et al., 2014*) and

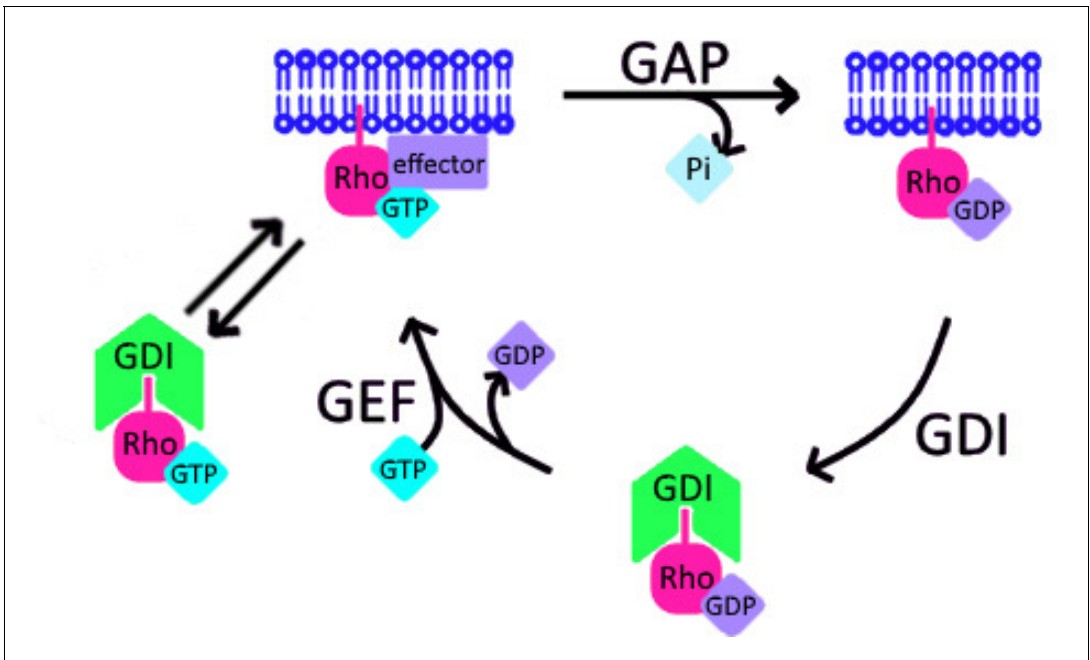

**Figure 10.** Schematic of proposed update to RhoGTPase cycle. We propose that in addition to the canonical GTPase cycle, GDI can extract active GTPase from the plasma membrane. Based on evidence that GTPase:GDI binding prevents GTP hydrolysis and nucleotide exchange (*Hart et al., 1992*; *Ueda et al., 2001*), active GTPase extracted by GDI would still be active upon its release back into the plasma membrane.
The online version of this article includes the following figure supplement(s) for figure 10:

**Figure supplement 1.** Schematic of RhoGDI's role in RhoGTPase zone definition around wounds.

modeling (*Holmes et al., 2016*) studies not only indicate that different isoforms of protein kinases C act at different regions around cell wounds, but also have different impacts on Rho and Cdc42 activity zones. When taken with the results of this current study, a picture emerges in which a high level of precision in GTPase patterning is achieved by a complex regulatory network comprising kinases, GEFs, GAPs, and GDI.

# Materials and methods

## Plasmids

The active RhoGTPase probes, mRFP-wGBD, eGFP-wGBD, eGFP-2xrGBD, BFP-2xrGBD and mRFP-2xrGBD in pCS2+ were generated as previously described (*Sokac et al., 2003*; *Benink and Bement, 2005*; *Davenport et al., 2016*). mCh-Rho, mCh-Rac, mCh-Cdc42 (*Benink, 2005*), untagged Rho, Rac, and Cdc42 (wild-type (WT) and constitutively-active (G14/2V)) in pCS2+ were made as previously described (*Benink and Bement, 2005*). For expression and purification from *E. coli*, codon-optimized Cdc42 and Rho (Eurofins Genomics Germany GmbH, Ebersberg, Deutschland) were subcloned into a pETMz2 vector via Gibson assembly cloning (*Gibson et al., 2009*), with primers GTPase(GeneStrand)fwd and -rev and pETfwd and -rev (all primer sequences in *Supplementary file 2*). A pentaglycine for sortase-mediated labeling was added onto the 5' of the GTPases. Constitutively-active Cdc42 Q61L and Rho Q63L mutants were generated by Quickchange mutagenesis with primers Cdc42(Q61L)fwd and -rev and Rho(Q63L)fwd and -rev, respectively.

*X. laevis* IT-Cdc42 in pCS2+ was generated according to *Bendezú et al. (2015)*: a linker - SGGSACSGPPG- was cloned into Cdc42 after Q134. The linker encodes for *BamH1* and *Asc1* restriction sites for digestion and insertion of GFP into the linker region. The 5' end of Cdc42 was amplified with primers Cdc42(1) and Cdc42(2); the 3' end was amplified separately with primers Cdc42(3) and Cdc42(4). The two products were joined by PCR stitching with primers Cdc42(1) and Cdc42(4). The single product was digested with *EcoR1* and *Xho1* and ligated into pCS2+. The resulting construct was mutated by Quickchange with primers pCS2+-Cdc42(1) and pCS2+-Cdc42(2) to remove the *BamH1* restriction site upstream of the insertion in the multiple cloning site. eGFP was amplified from eGFP-wGBD with primers eGFP(1) and eGFP(2). Both the Quickchanged construct and eGFP were digested with *BamH1* and *Asc1*, and eGFP was ligated into the linker region internal to the Cdc42 coding sequence. *X. laevis* Rho and Rac were similarly tagged internally after residues Q136 and L134, respectively.

To make C3-insensitive mCh-Rho and IT-Rho, constructs were Quickchanged to N41V using primers Rho(N41V)fwd and -rev (*Sekine et al., 1989*).

*X. laevis* RhoGDI Clone ID:7010361 (GE-Healthcare Dharmacon, Lafayette, CO) was subcloned into pCS2+ with *Cla1* and *Xho1*, and into N'3xGFP and N'Halo (Promega, Madison, WI) pCS2+ with *BspE1* and *Xho1*. A FLAG-tag was added by PCR onto the 5' of RhoGDI, and the product was subcloned into pFast-Bac1 with *Cla1* and *Not1*. The following mutations were made by Quickchange mutagenesis to untagged and N'3xGFP RhoGDI in pCS2+: E158/9A, E158/9Q, D40A, D40N, D180A and D180N (*Dransart et al., 2005*). Mutant RhoGDI 8(A) had the first eight charged amino acids to mutated to alanine (D3/5, E11-13, E15-17A) by sequential PCR (*Ueyama et al., 2013*). The 3' end of RhoGDI was amplified with primers 8(A)F1 and R1. The product was amplified and added to at its 5' end with primers 8(A)F2 and R1, and for a third time with 8(A)F3 and R1. The final PCR product was subcloned into pCS2+ by Infusion PCR (Takara Bio, Kusatsu, Japan). For subcloning into N'3xGFP-pCS2+, the third PCR from above was repeated with 8(A)F4 and R2. Mutant RhoGDI helix replacement (HR) had alpha helix D39-Q48 replaced with a glycine linker GGGGSGGGGS. This was done by sequential PCR as described above with four rounds of PCR: HR1 and R1, HR2 and R1, HR3 and R1, then either HR4 and R1 for subcloning into pCS2+ or HR5 and R2 for subcloning into N'3xGFP-pCS2+. RhoGDI mutant Δ51–199 was generated by adding a stop codon after L50 by Quickchange mutagenesis. To make RhoGDI Δ1–19, the 3' end of RhoGDI was amplified with primers (-)20 F1 and R for subcloning by infusion into pCS2+, and primers (-)20 F2 and R for N'3xGFP-pCS2+. Primers (−55) F1, F2 and R were used to generate RhoGDI Δ1–54 as described above (*Hoffman et al., 2000*; *Ueyama et al., 2013*). For expression and purification in *E. coli*, *X. laevis* RhoGDI WT and E158/9Q were subcloned into a pGEX-6P-2 vector via Gibson assembly cloning (*Gibson et al., 2009*) using XlRhoGDIfwd and -rev. A cysteine for labeling was added by PCR onto the 5' of

RhoGDI. Bovine RhoGDI1 in pGEX-6P was a gift of Dr. Tomotaka Komori. E163/4Q mutation was made by Quickchange mutagenesis with the E163/4Qfwd and -rev primers. Mutant bovine RhoGDI Δ1–22 and Δ1–59 were subcloned with BamHI and NotI into a pGEX-6P-2 with Δ1-22fwd and -rev and Δ1-59fwd and -rev, respectively. Mutant bovine RhoGDI HR was generated via Gibson assembly cloning (*Gibson et al., 2009*), with primers HRfwd and -rev and pGEXHRfwd and -rev, for amplification of RhoGDI and the pFASTBacH10 vector, respectively. RabGGTase two beta in a pGATEV vector was kindly provided by Dr. Konstantin Gavriljuk.

## Expression and purification of recombinant protein from *E. coli* for sortase-labeling

Rosetta(DE3) chemically competent *E. coli* cells were transformed with WT or mutant RhoGDIs, induced with 250 µM IPTG and incubated at 18°C ON. Bacteria cells were harvested, centrifuged at 4000xg for 20 min, and pellets flash frozen in liquid nitrogen and stored at −80°C. Frozen pellets were resuspended in a 3x volume of lysis buffer (50 mM KPi pH 8, 400 mM KCl, 1 mM EDTA, 5 mM βME, 1 mM PMSF, 1 mM benzamidine) and lysed with a high pressure homogenizer at 4°C. Lysate was clarified by centrifugation at 100,000xg for 1 hr and applied to a glutathione sepharose four fast flow column bed (GE-Healthcare, Chicago, IL) equilibrated with wash buffer (50 mM KPi pH 8, 400 mM KCl, 1 mM EDTA, 5 mM βME, 1 mM benzamidine). The column was washed with wash buffer, and protein was eluted with elution buffer (50 mM KPi pH 8, 400 mM KCl, 1 mM EDTA, 5 mM βME, 1 mM benzamidine, 10 mM reduced L-glutathione). Peak fractions were pooled, protein concentration was estimated with Bradford assay (Bio-Rad Laboratories, Inc, Hercules, CA), and PreScission protease was added at 1:30. After ON incubation on ice, the sample was concentrated using 5,000 MWCO Vivaspin15R centrifugal concentrators (Sartorius AG, Göttingen, Germany), buffer exchanged in wash buffer on a HiPrep 26/10 desalting column (GE-Healthcare) and recirculated on the same glutathione sepharose four fast flow column bed re-equilibrated in wash buffer. Flow-through was collected, concentrated, spun down and gel filtered on a HiLoad Superdex 75 pg column (GE-Healthcare) in storage buffer (20 mM HEPES pH 7.5, 150 mM KCl, 0.5 mM TCEP, 20% Glycerol). Peak fractions were pooled, concentrated, flash frozen in liquid nitrogen and stored at −80°C. Protein purification and purity were determined by Coomassie stain of 12% SDS-PAGE, protein concentration measuring absorbance at 280 nm. WT and mutant GTPases were expressed and purified similarly to GDIs with the following differences: (1) L21(DE3) chemically competent *E. coli* cells were used and proteins were expressed with 1 mM IPTG at 37°C for 4 hr; (2) the affinity step was performed on HiTrap Chelating HP columns loaded with cobalt and equilibrated in 50 mM HEPES pH 7.5, 50 mM NaCl, 5 mM MgCl2, 0.5 mM βME, 100 µM ATP and 100 µM GDP/GTP; (3) protein were gel filtered in storage buffer (50 mM HEPES pH 7.5, 50 mM NaCl, 2 mM MgCl2, 2 mM DTT, 20% Glycerol). RabGTTase Beta was expressed and purified as described before (*Gavriljuk et al., 2013*). Before each membrane extraction reaction, nucleotide bound to Cdc42 G12V was exchange to GTPγS, incubating the protein with 10-fold excess EDTA and GTPγS for 30 min on ice. The new nucleotide state was stabilized by adding 20-fold excess MgCl2. Exchange of nucleotide was confirmed by reversed phase chromatography using a C18 column under isocratic conditions (50 mM potassium phosphate pH 6.6, 10 mM tetrabutylammonium bromide, 16% (v/v) ACN) as described previously (*Müller et al., 2010*). Nucleotide peaks were quantified spectrometrically measuring absorbance at 254 nm.

## Cy3-labeling and in vitro prenylation of RhoGTPases

RhoGTPases were labeled at the N-t with Cy3 using a sortase-mediated reaction and in vitro prenylated as previously described (*Gavriljuk et al., 2013*; *Popp et al., 2007*). In brief, RhoGTPases were incubated with sortase and Cy3 N-t labeled LPETGG peptide at 3:1:15 ratio in labeling buffer (Tris pH 8.0, 150 mM KCl, 6 µM CaCl2, 0.5 mM TCEP) and incubated ON at 16°C. The entire reaction was mixed with geranylgeranyltransferase type one and geranylgeranyl diphosphate at 10:1:30 ratio in prenylation buffer (50 mM HEPES pH 7.5, 50 mM NaCl, 2 mM MgCl2, 2 mM DTT, 30 µM GDP/GTP, 2% CHAPS), and incubated ON on a rotating mixer at 4°C. The sample was spun in a TLA-100 rotor (Beckman Coulter, Brea, CA) at 80,000 rpm for 30 min at 4°C and gel filtered on a HiLoad Superdex 75 pg column (GE-Healthcare) equilibrated with prenylation buffer with 0.5% CHAPS. Peak fractions were pooled, concentrated using 5,000 MWCO Vivaspin4 centrifugal concentrators (Sartorius AG)

and buffer exchanged in prenylation buffer without CHAPS on a NAP-5 column (GE-Healthcare). Residual detergent was removed by Pierce Detergent Removal Spin Column (Thermo Fisher, Carlsbad, CA). After sortase-mediated labeling with Cy3, unprenylated proteins were directly spun down and gel filtered in absence of CHAPS.

## Expression and purification of recombinant protein from insect cells

DH10Bac-competent *E. coli* (Thermo Fisher) were transformed with FLAG-WT RhoGDI or E158/9Q in pFast-Bac1 and positive clones were identified by blue/white screening. Bacmid was purified and transfected into Sf9 cells using Cellfectin II reagent (Thermo Fisher). High-expressing clones were identified and baculovirus was generated for two additional passages. Sf9 cells, $22 \times 10^6$ per 15 cm plate, were infected with high-titer baculovirus and incubated 27°C for 72 hr. Sf9 cells were harvested, centrifuged at 500xg for 5 min, and pellets were stored at −80°C.

Frozen pellets were resuspended in a 5x volume of solubilization buffer (1xPBS pH 7.5, 1% Triton X-100, 0.5 µg/mL leupeptin, 0.5 µg/mL aprotinin, 0.5 µg/mL Pepstatin A, 40 µg/mL PMSF, 100 µg/mL benzamidine, 0.5 µg/mL E64) and incubated at 4°C with end-over-mixing for 1 hr. Lysate was clarified by centrifugation at 21,000xg for 15 min and applied to an anti-FLAG M2 agarose column bed (MilliporeSigma, Burlington, MA). The column was washed 3x with wash buffer (1xPBS, 0.5 µg/mL leupeptin, 0.5 µg/mL aprotinin, 0.5 µg/mL Pepstatin A, 40 µg/mL PMSF, 100 µg/mL benzamidine, 0.5 µg/mL E64). A buffer exchange was performed with 1:1 wash buffer:HEPES (25 mM HEPES pH 7.5, 100 mM KCl), and the column washed 2x with HEPES. Protein was eluted with 1M Arginine pH 4.4 into an equal volume of collection buffer (50 mM HEPES pH 7.5, 200 mM KCl). Fractions were analyzed by coomassie stain of a 12% SDS-PAGE. Peak fractions were pooled and concentrated using a 10 MW Amicon Ultra-15 Centrifugal filter (MilliporeSigma). A buffer exchange was performed during concentrating with HEPES such that the final Arginine concentration was less than 1 mM. Protein purification and purity was determined by comparison to a BSA standard curve by Coomassie stain of a 12% SDS-PAGE.

## Oocyte collection and preparation

Ovarian tissue was harvested from adult *X. laevis* via surgical procedures approved by the University of Wisconsin-Madison Institutional Animal Care and Use Committee. Oocytes were stored in 1x modified Barth's solution (88 mM NaCl, 1 mM KCl, 2.4 mM NaHCO₃, 0.82 mM MgSO₄, 0.33 mM NaNO₃, 0.41 mM CaCl₂, 10 mM HEPES, pH 7.4) with 100 µg/mL gentamicin sulfate, 6 µg/mL tetracycline and 25 µg/mL ampicillin at 16°C. Prior to manual defolliculation with forceps, oocytes were treated with 8 mg/mL type I collagenase (Life Technologies, Grand Island, NY) in 1x modified Barth's solution for 1 hr at 16°C on an orbital shaker.

## mRNA preparation

mRNA was generated in vitro using the mMessage mMachine SP6 transcription kit (Thermo Fisher) and purified using the RNeasy Mini Kit (Qiagen, Hilden, Germany). Transcript size was verified on a 1% agarose/formaldehyde denaturing gel relative to the Millennium Marker (Life Technologies) RNA molecular weight standard.

## Oocyte microinjection

Oocytes were microinjected with a 40 nL injection volume using a p-100 microinjector (Harvard Apparatus, Holliston, MA). mRNA encoding probes for active Rho (2xrGBD) and active Cdc42 (wGBD) were injected at a final needle concentration of 30 µg/mL and 100 µg/mL, respectively. IT-Rho, Rac and Cdc42 were each injected at a final needle concentration of 125 µg/mL, with 63 µg/mL of WT RhoGDI to stabilize the exogenous GTPase and maintain stoichiometric ratio of GTPase:GDI (*Boulter et al., 2010*). mCh-Rho, Rac and Cdc42 were each injected at a final needle concentration of 125 µg/mL. mRNA encoding Abr was injected at a final needle concentration of 25–500 µg/mL. 3xGFP-WT GDI and mutants at 333 µg/mL, untagged RhoGDI at 300 µg/mL, Halo-WT GDI and mutants at 200 µg/mL, Cdc42 G12V or Q61L at 28 µg/mL, and untagged GDI E158/9Q at 1.5 mg/mL. For purified protein, Cy3-Rho and Cy3-Cdc42, in vitro prenylated and complexed with RhoGDI, were injected at a final needle concentration of 4.56 µM. C3 exotransferase was injected at a final needle concentration of 1.1 µg/mL in 1 mM DTT and WT GDI at 3.5–114 µM for the standard curve.

For wounding experiments, all mRNA was injected 20–24 hr before imaging, and purified protein was injected at least 2 hr before imaging, except for C3 which was injected 30 min prior to imaging. For imaging cortical granule exocytosis, oocytes were injected 16 hr before imaging and matured ON in progesterone. Two-cell embryos were microinjected with a 5 nL injection volume at a final needle concentration of 167 µg/mL for IT-Rho and IT-Cdc42 mRNA, and 18.24 µM Cy3-Rho and Cy3-Cdc42.

## Quantification of IT-Rho expression level in *X. laevis* oocyte

Whole cell lysates (WCL) were generated from oocytes with or without IT-Rho expression. Oocytes were washed 3x with wash buffer (10 mM imidazole, 50 mM KCl, 2.5 mM $MgCl_2$, 1 mM EGTA, 10 mM EDTA). 5 µL/oocyte of lysis buffer (wash buffer supplemented with 1 mM DTT, 0.5% TritonX-100, 50 µg/mL E64, 4 mM Peflabloc, 60 µg/mL chymostatin, 5 µg/mL leupeptin, 1 µg/mL pepstatin, 4 µg/mL aprotinin, 2 µM calpeptin, 2 µM ALLN) was added and oocytes were homogenized with a 200 µL pipet tip. Homogenates was added to a 3/16 × 25/32 Ultra-clear centrifuge tube (Beckman Coulter, Ref:344718), centrifuged at max speed for 5 min at 4°C and the cytoplasmic fractions were extracted by piercing the side of the tube with a syringe. The cytoplasmic fractions were transferred to 1.5 mL tubes, centrifuged at max speed for 5 min at 4°C, and the cytoplasmic fractions was isolated using a pipet. 6X LSB was added, samples were boiled for 10 min and separated by 12% SDS-PAGE. IT-Rho expression was determined by western blot of the equivalent of 1 or two oocyte volumes (lanes specified) alongside a standard curve of purified GFP-UtrCH 261, generously provided by Kevin Sonnemann. The blot was probed with anti-GFP (B-2) (1:1000, sc-9996, Santa Cruz, Dallas, TX) primary and goat anti-mouse IR-Dye 800CW (1:10,000, 926–32210, LI-COR, Lincoln, NE) secondary antibodies. Visualization was achieved with a LI-COR Odyssey Fc imaging system.

## Purification of recombinant protein from *E. coli* for antibody purification

BL21 pLysS cells (Thermo Fisher) were transformed with GST-RhoGDI in pGEX6p.1. A positive clone was used to inoculate 12 mL of lysogeny broth (LB) supplemented with 25 µg/mL ampicillin and cultured ON. The 12 mL culture was added to 1L of LB with ampicillin and shaken at 37°C until OD600 ~0.6. The culture was induced by adding a final concentration of 0.1 mM IPTG and shaken at 37°C for 2 hr. BL21 pLysS cells were pelleted at 5300 rpm for 10 min at 4°C, and the pellet resuspended in Buffer A (50 mM Tris-HCL, pH 7.6; 50 mM NaCl with 1 mM DTT in PBS). Pellets were stored at −80°C. Pellets were thawed at room temperature to promote cell lysis. Triton X-100 was added to a final concentration of 0.6%, PMSF at 500 uM, lysoszyme at 1 mM in 10 mM Tris pH 8.0, 400 µM Peflabloc, 1 µg/mL aprotinin, 1 µg/mL leupeptin. Solubilate was incubated at RT for 30 min, DNAse1 was added to a final concentration of 10 ug/mL, incubated again for at RT for 30 min, and centrifuged at 16,000xg for 10 min at 4°C. The supernatant was collected and exposed to a column containing glutathione-sepharaose 4B (MilliporeSigma). The column was washed 5x with Buffer A and the protein eluted with 20 mM Tris, pH 8.0, 20 mM glutathione, 400 µM Peflabloc, 1.25 µg/mL aprotinin, 14.25 µg/mL leupeptin, 0.25 mM E-64, 0.5 mM PMSF. Protein concentration was determined by Coomassie stain of a 12% SDS-PAGE alongside a BSA standard curve.

## Antibody generation and purification

FLAG-RhoGDI purified from Sf9 cells was used as an antigen for antibody production in rabbits (Covance, Princeton, NJ). The serum was heat-inactivated at 56°C for 30 min, diluted 1:1 in 20 mM Tris, pH 7.5, and filtered through a 0.22 µm syringe. The diluted, filtered serum was loaded onto a column containing GST-GDI coupled to Affi-Gel 15, to minimize antibody cross-reactivity to the FLAG-tag on the antigen. The column was washed 20x with 20 mM Tris, pH 7.5 and 20x with 20 mM Tris, pH 7.5, 500 mM NaCl. Antibody was first eluted with 100 mM glycine, pH 2.5 into 1M Tris, pH 8.8 for neutralization. The column was washed 20x with 20 mM Tris, pH 8.8. Antibody remaining on the column was eluted with 100 mM Triethylamine, pH 11.5 into concentrated HCl and 1M Tris, pH 7.5 for neutralization. The concentration of each fraction was determined by $A_{280.}$ The peak antibody fractions were pooled, dialyzed against PBS (2 × 2L) ON, and concentrated using a 100 K MW Amicon Ultra-15 Centrifugal filter (MilliporeSigma). An equal volume of glycerol was added and stored at −20°C. Antibody specificity was determined by western blotting of purified protein, *X. laevis*

oocyte whole cell lysate (WCL, described above), WCL of oocytes overexpressing GDI and WCL of oocytes expressing 3xGFP-GDI, separated by 12% SDS-PAGE. The blot was probed with anti-GDI (1:1000) primary and Alexa-Fluor 680 goat anti-rabbit (1:10,000, A21076, Molecular Probes, Eugene, OR) secondary antibodies. Visualization was achieved with a LI-COR Odyssey Fc imaging system.

## Fixing and staining of wounded oocytes

Oocytes were wounded, allowed to heal for 2–3 min and fixed for 2 hr in 10 mM EGTA, 100 mM KCl, 3 mM MgCl$_2$, 10 mM HEPES, 150 mM sucrose (pH 7.6), 4% PFA, 0.1% glutaraldehyde, 0.1% Triton X-100. Fixed oocytes were washed 5x in TBSN/BSA (5 mg/mL BSA in 1xTBS containing 0.1% NP-40). Oocytes were bisected and blocked in TBSN/BSA for 4 hr at 4°C. Oocytes were stained with rabbit α-RhoGDI at 1:1000 in TBSN/BSA for 12 hr, washed 5x in TBSN/BSA over 12 hr, stained with chicken α-rabbit Alexa Fluor 647 (Invitrogen, Carlesbad, CA) at 1:10,000 for 12 hr in BSN/BSA at 4°C, and washed 5x in TBSN/BSA over 12 hr.

## Image acquisition, wounding and data analysis

Laser scanning confocal microscopy was performed using a Nikon Eclipse Ti inverted microscope with a Prairie Point Scanner confocal system (Bruker, Middleton, WI). The microscope was fitted with a 440 nm dye laser pumped by a MicroPoint 337 nm nitrogen laser (Andor, South Windsor, CT) for wounding. Brightest-point projections, measurements of fluorescence intensities, area and distances were made in FIJI (*Schindelin et al., 2012*). Bio-Formats Importer and De-Flicker plugins were used. Ring intensity corrected for background was calculated by quantifying the mean intensity of the ring and subtracting the mean intensity of the background. Total activity was calculated by multiplying the mean intensity of the zone (corrected for background) by the area of the zone, normalized for wound width. GraphPad Prism was used to plot quantifications and perform statistical analyses. An unpaired student's T-test with a 2-tailed distribution and unequal variance was used to compare two conditions, one-way ANOVA with a Tukey post hoc analysis was used to analyze more than two conditions. *p<0.05, **p<0.01, ***p<0.001, ****p<0.0001.

## Supported lipid bilayer assay

Small unilamellar vesicles and supported lipid bilayers were prepared with 100% 1,2-dioleoyl-sn-glycero-3-phosphocholine (18:1 DOPC; Avanti Polar Lipids, Inc, Alabaster, AL) as described before (*Hansen et al., 2019* ). 250µL Cy3 labeled GTPases were incubated on SLBs at 200nM final concentration until equilibrium was reached. A 0.5mm silicone tubing was attached drop-to-drop to the chamber with the equilibrated sample via a male luer connector (ibidi GmbH, Gräfelfing, Germany). After acquisition of few frames in absence of flow, the chamber was flushed at 10µL/sec with imaging buffer (20mM HEPES pH 7.0, 150mM KCl, 1.5mM MgCl$_2$, 0.5mM EGTA, 100µM GDP/GTP) alone or in presence of a GTPase solubilizer until baseline was reached. Oxygen scavenger system (1.25mg/mL glucose oxidase, 0.2 mg/mL catalase, 400 mg/mL glucose) was added fresh to each sample and buffer before imaging. TIRF was performed on a Nikon Eclipse Ti inverted microscope with a VisiScope TIRF-FRAP Cell Explorer system (Visitron Systems GmbH, Puchheim, Germany) using a 60× Apo TIRF oil-immersion objective (1.49 N.A.). Cy3-labeled proteins were excited with a 561-nm laser line, excitation light was passed through a ET-561nm Laser Bandpass Set (Chroma Technology Corporation, Bellows Falls, VT) before illuminating the sample. Fluorescence emission was detected on a Evolve 512 Delta EMCCD camera (Teledyne Photometrics, Tucson, AZ). Measurements of fluorescence intensities were made in FIJI (*Schindelin et al., 2012*). Plot Z-axis profile tool and Bio-Formats Importer plugins were used. Fluorescence intensity was corrected for background. To display multiple curves on the same graph, data from different experiments were aligned using the overshoot signal occurring after the flow was started and normalized dividing by the maximum intensity. Data from wash off experiments were fitted with a one component exponential decay function ($y = y_0 + A\ e^{-\lambda\ x}$; $y_0$= y offset, $A$=amplitude, $\lambda$=exponential decay constant), choosing a fitting range that did not include the initial overshoot. This was possible because a monoexponential function can be fitted to a range of the data set without affecting the $\lambda$ value obtained. $K_{off}$ titration curves were fitted with a hyperbolic function ($y = y_0 + \frac{\lambda_{max}\ x}{K_d + x}$). Origin Pro (OriginLab Corporation, Northampton, MA) was used to analyze data, plot quantifications and perform statistical analyses.

An unpaired student's T-test with a 2-tailed distribution and equal variance was used to compare two conditions. **p<0.01, ***p<0.001, ****p<0.0001.

## Acknowledgements

This work was supported by National Institutes of Health Grant GM52932 to WMB, a Dr. Stanley and Dr. Eva Lurie Weinreb Fellowship to AEG and HSFP CDA00070/2017-2 to PB. IV is supported by the MaxSynBio Consortium, which is jointly funded by the Federal Ministry of Education and Research of Germany and the Max Planck Society. We acknowledge National Institutes Health Grant R44 MH065724 to LOCI at UW-Madison. We are also grateful to Dr. Kevin Sonnemann (UW-Madison) for purified GFP-UtrCH 261 protein, the Sherer lab (UW-Madison) for an anti-GFP antibody, Nathalie Bleimling and Dr. Amrita Rai (MPI-Dortmund) for help with the nucleotide exchange and HPLC quantification, Prof. Roger S Goody and Dr. Ingrid Vetter (MPI-Dortmund) for useful discussion, and both our labs for their continued input.

## Additional information

### Funding

| Funder | Grant reference number | Author |
| --- | --- | --- |
| National Institutes of Health | GM52932 | William M Bement |
| University of Wisconsin-Madison | Dr. Stanley and Dr. Eva Lurie Weinreb Fellowship | Adriana E Golding |
| Human Frontier Science Program | HSFP CDA00070-2017-2 | Peter Bieling Ilaria Visco |
| Max Planck Society | MaxSynBio Consortium | Ilaria Visco Peter Bieling |
| Deutsche Forschungsgemeinschaft | 399893760 | Peter Bieling Ilaria Visco |

The funders had no role in study design, data collection and interpretation, or the decision to submit the work for publication.

### Author contributions

Adriana E Golding, Ilaria Visco, Conceptualization, Data curation, Formal analysis, Validation, Investigation, Visualization, Methodology, Writing—original draft, Writing—review and editing; Peter Bieling, Conceptualization, Resources, Supervision, Funding acquisition, Methodology, Writing—original draft, Project administration, Writing—review and editing; William M Bement, Conceptualization, Resources, Formal analysis, Supervision, Funding acquisition, Investigation, Methodology, Writing—original draft, Project administration, Writing—review and editing

### Author ORCIDs

Adriana E Golding https://orcid.org/0000-0003-4305-0764
Ilaria Visco https://orcid.org/0000-0003-3753-6434
Peter Bieling https://orcid.org/0000-0002-7458-4358

### Ethics

Animal experimentation: The University of Wisconsin-Madison Animal Care and Use Committee has reviewed and approved all of the experiments performed for this study, outlined in protocol G005386-RO1.

### Decision letter and Author response

Decision letter https://doi.org/10.7554/eLife.50471.SA1
Author response https://doi.org/10.7554/eLife.50471.SA2

## Additional files

### Supplementary files

• Supplementary file 1. Apparent affinities and maximum velocities determined by hyperbolic fits to the RhoGDI extraction rates of membrane-bound RhoGTPases.

• Supplementary file 2. Primers.

• Transparent reporting form

### Data availability

All quantifications made in this study are included as source data files by figure.

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
