## [Decision Letter]

**Acceptance summary:**

This work revisits an old but still not well understood aspect of cell biology, namely, the role of Rho-GDIs in regulating Rho GTPase signaling. GDIs are relatively understudied compared to GEFs and GAPs and this work helps us better understand how the Rho cycle is regulated in time and space. There has long been a debate about the role of Rho-GDIs in regulating Rho family GTPases. Classically, the model has been that Rho-GDIs extract inactive Rho proteins from membranes and sequester them in the cytosol. The authors here challenge that model, and present a series of straightforward experiments in *Xenopus* and in vitro that indicate a more complex role for these proteins. Using new tools such as internally tagged small GTPases and GDI mutants that cannot extract small GTPases, the authors find that i) inactive GTPases also reside at the plasma membrane, and not just in a soluble pool bound to Rho-GDI; ii) that GDI differentially suppress active Cdc42 versus Rho; that GDI can extract active GTPases from the membrane; and that, in extraction deficient mutants, levels of active Cdc42 are elevated in vivo, confirming a physiological role, at least in this system. As a result of these experiments, we will need to revise the standard textbook models of Rho GTPase cycling, which, in turn, will make us rethink how cells regulate shape and motility in response to external signals and to internal programs.

**Decision letter after peer review:**

Thank you for submitting your article "Extraction of active RhoGTPases by RhoGDI regulates spatiotemporal patterning of RhoGTPases" for consideration by *eLife*. Your article has been reviewed by three peer reviewers, including Jonathan Chernoff as the Reviewing Editor and Reviewer #1, and the evaluation has been overseen by Anna Akhmanova as the Senior Editor.

The reviewers have discussed the reviews with one another and the Reviewing Editor has drafted this decision to help you prepare a revised submission.

Summary:

This is an interesting and well-done piece of work that revisits the potential role of Rho-GDIs in regulating Rho GTPase signaling. GDIs are relatively understudied compared to GEFs and GAPs and this work could help us better understand how the Rho cycle is regulated in time and space.

There has long been a debate about the role of Rho-GDIs in regulating Rho family GTPases. Classically, the model has been that Rho-GDIs extract inactive Rho proteins from membranes and sequester them in the cytosol. The authors here challenge that model, and present a series of straightforward experiments in *Xenopus* and in vitro that indicate a more complex role for these proteins.

Using new tools such as internally tagged small GTPases and GDI mutants that cannot extract small GTPases, the authors find that i) inactive GTPases also reside at the plasma membrane, and not just in a soluble pool bound to Rho-GDI; ii) that GDI differentially suppress active Cdc42 versus Rho; that GDI can extract active GTPases from the membrane; and that, in extraction deficient mutants, levels of active Cdc42 are elevated in vivo, confirming a physiological role, at least in this system.

Essential revisions:

One general criticism of this work is that the results, though convincing, seem somewhat oversold. That is, there is already evidence that Rho-GDIs can interact with active GTPases, as shown in several works from the Cerione lab (e.g., Nomanbhoy and Cerione 1996), so one of the main claims in this manuscript ought to be toned down. That is particular so because, even in this work, the degree to which Rho-GDIs extract active GTPases is far less than that of inactive GTPases. Therefore, the overall model represents more of an important tweak than a revolution in modeling Rho cycling dynamics. However, given the importance of such dynamics to many essential cellular processes, even a key tweak is of great value to the field.

1) Assuming that appropriate antibodies are available, In Figures 1 and 2, it would be useful to get a sense of relative levels of exogenous versus endogenous Rho and Cdc42, respectively. The authors mention potential issues related to overexpression and it would be good to know how much exogenous GTPase is being introduced into the oocytes.

2) Given that Q61L/Q63 forms of these GTPases are unable to cleave GTP, yet still (by the author's model) can be extracted by Rho-GDI, I would like to see a simple assay showing that such complexes (e.g. Cdc42 L61:Rho-GDI) can be isolated from the cytosol by IP. If this experiment isn't technically feasible, I would accept their current evidence as sufficient.

3) To establish the relative localization of Rho-GDI and the GTPases (active pools), the authors should perform line-scans on wound sites like in Figure 4 and report the distance between the peaks for a population of observed cells/wounds.

4) In Figure 1—figure supplement 1, the authors should discuss what they see. I see that the mCherry-Cdc42 zone is a bit larger than the active Cdc42 zone. For RhoA, I am surprised because I cannot see a clear difference by eye. The line-scan is meaningless because of the low SNR. Please discuss this point. One would need a fully orthogonal method to really decide which of the fusion protein is functional. Immunostaining would be the method of choice, but it is known that antibodies (raised to the C-term hypervariable region) are often of only low quality. I do not have a problem that IT-GTPase constructs are good markers of Rho GTPase localization.

5) Figure 5C: The authors make a strong point that GDI affects Cdc42 more robustly than RhoA. To support this point, a statistical analysis will be needed.

6) The authors use RhoA and Cdc42 (Q63/Q61L) mutants in their biochemical assays. They mention that G12/G14V RhoA and Cdc42 mutants work equally well. I would show that data. The work of Michaelson (Michaelson et al., 2001) clearly shows that G12/G14V Cdc42/Rho mutants retain GDI interactions in cells, whereas Q61/Q63L Cdc42/Rho mutants do not. Experiments with a RhoA biosensor shows that Q63L RhoA biosensors are inactivatable with GDI overexpression, whereas this is not the case with the Q63L mutation (Pertz et al., 2006). I would expect that the G12/G14V Cdc42/Rho mutants interact in a different fashion with GDI than their Q61/Q63L counterparts. These issues do not necessarily require new experiments, but will require adjustments to the Discussion.

7) Given the provocative possibility that there might be different molecular requirement for different spatio-temporal patterns (e.g. Rho and Cdc42 zones), I would strongly suggest that the authors perform an experiment with photoactivatable biosensors to measure the directional GTPase flux in response to the GDI perturbations (overexpression of WT and mutant). This would add greatly to the paper.

8) The paper would benefit from a discussion about the complexity of spatio-temporal Rho GTPase signaling, to address the following "big picture" points: Where is the field going? What needs to be done? I would also highlight the complexity of these signaling networks, with the requirement of the zone initiation/maintenance being different for Rho and Cdc42 (as depicted in Figure 10—figure supplement 1). Further it would be good to mention that Rho GTPase -GDI interactions can be regulated by phosphorylation: both at the level of the GTPase (RhoA is phosphorylated by PKA at its C-term – this regulates GDI interaction PMID: 8599934), and at the level of GDI (by phosphorylation by PAK PMID: 15225553, or Src PMID: 16943322). These phosphorylation events might provide feedback regulation (by Rho GTPase or adhesion signaling) to spatially fine tune the system.

---

## [Author Response]

Essential revisions:One general criticism of this work is that the results, though convincing, seem somewhat oversold. That is, there is already evidence that Rho-GDIs can interact with active GTPases, as shown in several works from the Cerione lab (e.g., Nomanbhoy and Cerione 1996), so one of the main claims in this manuscript ought to be toned down. That is particular so because, even in this work, the degree to which Rho-GDIs extract active GTPases is far less than that of inactive GTPases. Therefore, the overall model represents more of an important tweak than a revolution in modeling Rho cycling dynamics. However, given the importance of such dynamics to many essential cellular processes, even a key tweak is of great value to the field.

This is a good point. Our intention was to point out that in the textbook model of RhoGTPase cycling, GDI is thought to extract GTPases solely after GAP-dependent inactivation. However, the manner in which the original introduction was written might lead people to believe that the idea of active GTPase extraction by GDI had never been previously suggested. We have therefore rewritten the Introduction to make it clear that some earlier data shows that GDI can interact with active GTPases and that others previously suggested the possibility that this interaction might result in extraction. Specifically, we have added the following sentence to the Introduction: "However, work by several labs has shown that GDIs bind both inactive and active GTPases with relatively high affinity in vitro(Hancock and Hall, 1993; Hart et al., 1992; Nomanbhoy and Cerione, 1996; Tnimov et al., 2012), leading to the suggestion that GDIs interact with active as well as inactive GTPases in vivo". To further clarify this point, we have added the following to the end of the Introduction: "Using these tools, we identify co-existing pools of active and inactive GTPases associated with the plasma membrane and provide additional evidence that GDI can extract both inactive and active GTPases in vitro, as suggested by previous steady-state affinity measurements (Hancock and Hall, 1993; Leonard et al., 1992; Nomanbhoy and Cerione, 1996; Tnimov et al., 2012)".

As a further means to avoid "overselling", we removed the clause "While these findings might seem heretical" from the Discussion paragraph that now begins "This finding has the virtue…".

1) Assuming that appropriate antibodies are available, In Figures 1 and 2, it would be useful to get a sense of relative levels of exogenous versus endogenous Rho and Cdc42, respectively. The authors mention potential issues related to overexpression and it would be good to know how much exogenous GTPase is being introduced into the oocytes.

This is another good point. We have calculated the relative level of exogenous vs. endogenous Rho and Cdc42 for the Cy3-labeled GTPases using the proteomic data from Wuhr et al., 2014, for *X. laevis* eggs. For Cy3-Rho, the concentration used in this study results in a 41% increase above endogenous levels. For Cy3-Cdc42, the concentration used in this study results in a 52% increase above endogenous levels. Importantly, the microinjected Cy3-GTPases are bound to GDI, so there is presumably no danger of disrupting the GDI-GTPase balance, which can lead to GTPase aggregation and degradation (see Boulter et al., 2010). This information has been added to the Results section (paragraph three).

For the internally-tagged (IT) GTPases, more effort was needed. Unfortunately, available antibodies for Rho and Cdc42 do not work well in the *Xenopus* system. Instead, we used the following strategy: oocytes microinjected with mRNA encoding IT-Rho (GFP), at the same concentration used for imaging throughout the paper, were homogenized and whole cell lysates were separated by SDS-PAGE alongside standard curve of purified GFP-UtrCH 261 (Burkel et al., 2007). The blot was probed with an anti-GFP antibody. The results obtained showed that the concentration of IT-GFP-Rho mRNA microinjected in this study results in a 36% increase above endogenous Rho levels. As with the Cy3-labeled GTPases, the IT GTPases are co-expressed with low concentrations of GDI mRNA to avoid disrupting the GDI-GTPases balance. This information has been added to the Results (paragraph two) and Materials and methods (subsections “Oocyte microinjection” and “Quantification of IT-Rho expression level in *X. laevis*oocyte”).

2) Given that Q61L/Q63 forms of these GTPases are unable to cleave GTP, yet still (by the author's model) can be extracted by Rho-GDI, I would like to see a simple assay showing that such complexes (e.g. Cdc42 L61:Rho-GDI) can be isolated from the cytosol by IP. If this experiment isn't technically feasible, I would accept their current evidence as sufficient.

We agree with the reviewers that it would be ideal to provide additional evidence for the existence of soluble complexes between GDI and active GTPases in the cytosol. However, based on our biochemical understanding of the interaction between GDI and its GTPase clients, we would not anticipate these complexes to be sufficiently stable to allow for isolation via immunoprecipitation. As part of a separate project, we have investigated this aspect in biophysical detail. Both inactive and active GTPases interact with GDI with sub-micromolar affinity, indicating that complexes should exist at physiological protein concentrations. However, the dissociation rates of GDI from either inactive (GDP-bound) or active (GTP-bound, Q61L) Cdc42 differ by more than two orders of magnitude (see Author response image 1). GTP-bound Cdc42(Q61L) releases GDI with half-lives of less than 10s, showing that the complex should be expected to fall apart during the inevitable dilution and the repeated wash steps required for IP.

Nonetheless, we tried to address this point by microinjecting either Cy3-labeled Q61L Cdc42 or, as a negative control, Cy3-Cdc42 R66E (a mutant which does not interact with GDI). Cells were homogenized, mixed with purified FLAG-GDI, and coupled to a FLAG affinity resin for precipitation. Samples were separated by SDS-PAGE, transferred and blotted with a commercially obtained anti-Cy3 antibody. Unfortunately, the antibody recognized a number of proteins, most of which were too large to be GDI.

**Author response image 1. respfig1:** Dissociation of either GTP-loaded, Cy3-Cdc42 (Q61L) (yellow) or GDP-loaded Cy3-Cdc42 (WT) from Alexa647-RhoGDI after addition of excess unlabeled RhoGDI at t=0 as monitored by FRET..

3) To establish the relative localization of Rho-GDI and the GTPases (active pools), the authors should perform line-scans on wound sites like in Figure 4 and report the distance between the peaks for a population of observed cells/wounds.

This is a very useful suggestion. We have performed additional triple-label experiments and now display the information obtained from them in revised Figure 4A,B, which includes both a radially-averaged kymograph and quantification of the results obtained from multiple cells. The data show that the peak of GDI signal consistently lags behind the peak of Rho signal, which supports the notion that GDI may differentially regulate Cdc42 and Rho in vivo.

4) In Figure1—figure supplement 1, the authors should discuss what they see. I see that the mCherry-Cdc42 zone is a bit larger than the active Cdc42 zone. For RhoA, I am surprised because I cannot see a clear difference by eye. The line-scan is meaningless because of the low SNR. Please discuss this point. One would need a fully orthogonal method to really decide which of the fusion protein is functional. Immunostaining would be the method of choice, but it is known that antibodies (raised to the C-term hypervariable region) are often of only low quality. I do not have a problem that IT-GTPase constructs are good markers of Rho GTPase localization.

This is an important point to address. Put simply, relative to the IT-GTPases, the N-terminally FP-tagged GTPases show either very faint recruitment to wounds, very broad recruitment to wounds, or both. We have therefore added text to this effect to the Results: "Specifically, the zones of recruitment defined by amino-terminally tagged GTPases are much less focused and much less intense than those obtained with either the activity reporters or the internally tagged GTPases (Figure 1—figure supplement 1; see below for functional analysis)".

We have also increased the signal on the image in the supplemental figure to make it clearer how different it is from the IT-GTPase. In addition, to address this point, we have also included references that have reported that Rho tagged N-terminally with a fluorescent protein fails to concentrate at the cleavage furrow in human cells and added additional references to document that amino terminal-tagging of RhoGTPases with fluorescent proteins impairs their localization and function (Introduction paragraph four and Results paragraph one).

Most importantly, we have directly tested the ability of C3-insensitive, amino-terminally GFP-tagged Rho to rescue Rho activation at wounds in C3 injected cells. In contrast to the C3-insensitive IT-Rho, the N-terminally-tagged version failed to rescue Rho activity at wounds in C3-injected cells. These data have been added to Figure 2H, Figure 2—figure supplement 1, and are discussed in the Results section (paragraph five).

5) Figure 5C: The authors make a strong point that GDI affects Cdc42 more robustly than RhoA. To support this point, a statistical analysis will be needed.

We have performed and added statistical analyses that show that very low level (~15%) overexpression of both *Xenopus* (Figure 5C,D)) and bovine (Figure 5—figure supplement 2) RhoGDI significantly reduces Cdc42 activity but not Rho activity at wounds. These findings are described in the Results.

6) The authors use RhoA and Cdc42 (Q63/Q61L) mutants in their biochemical assays. They mention that G12/G14V RhoA and Cdc42 mutants work equally well. I would show that data. The work of Michaelson (Michaelson et al., 2001) clearly shows that G12/G14V Cdc42/Rho mutants retain GDI interactions in cells, whereas Q61/Q63L Cdc42/Rho mutants do not. Experiments with a RhoA biosensor shows that Q63L RhoA biosensors are inactivatable with GDI overexpression, whereas this is not the case with the Q63L mutation (Pertz et al., 2006). I would expect that the G12/G14V Cdc42/Rho mutants interact in a different fashion with GDI than their Q61/Q63L counterparts. These issues do not necessarily require new experiments, but will require adjustments to the Discussion.

We agree with the reviewer that these references suggest that GDI interacts differently with G12/14V and Q61/63L variants in cells. Whether this is due to i) these mutations differentially weakening GDI binding directly or ii) indirect effects resulting from the differences in nucleotide states these variants assume in the cytoplasm is not known. The former possibility motivated us to directly compare these two variants more carefully in additional in vitroexperiments.

Our previous data suggested that G12/14V and Q61/63L variants behave similarly in our membrane extraction assay. However, we knew that our G12/14V data was compromised by i) incomplete nucleotide exchange and ii) the continual hydrolysis of during the long overall duration of the experiment, resulting in a mix of nucleotide states. We have now improved the assay by exchanging the nucleotide directly and more completely before each membrane extraction reaction, thereby eliminating the confounding issue of mixed nucleotide states.

This new data confirms that active, GTPγS-loaded G12/14V GTPases are also potently extracted by RhoGDI. In fact, extraction is even more rapid compared to the corresponding Q61/Q63L variants. We draw two important conclusions from this data: i) extraction of active GTPases is a robust feature of GDI that we observe for distinct constitutively-active GTPase variants and among GDI orthologs, strengthening one of our central conclusions; ii) G12/14V and Q61/63L variants differ moderately in their ability to interact with GDI on membranes.

The latter point is important, because we find that our extraction-deficient GDI mutant (GDI-QQ) retains residual activity in extracting GTPγS-loaded G14V, but not GTP-loaded Q63L Rho. This offers an alternative interpretation of the differential effects of GDI-QQ overexpression on Rho versus Cdc42 activity around wounds. GDI-QQ might still be able to attenuate Rho, but not Cdc42 activity by direct extraction.

We have added the new data to Figure 7, Figure 8 and Figure 7—figure supplement 3 and discuss these findings now in detail in the Results and Discussion sections of the manuscript.

7) Given the provocative possibility that there might be different molecular requirement for different spatio-temporal patterns (e.g. Rho and Cdc42 zones), I would strongly suggest that the authors perform an experiment with photoactivatable biosensors to measure the directional GTPase flux in response to the GDI perturbations (overexpression of WT and mutant). This would add greatly to the paper.

While we would like to perform such experiments, they are far, far beyond the scope of the current work both in terms of added material and added time. With respect to added material, conducting flux experiments would add at least two new figures and another set of concepts to be digested by the readers. We believe that what is already contained in our manuscript represents a complete study both conceptually and in terms of the amount of information. The study in its current form contains extensive, parallel in vivoand in vitroexperiments, it reveals several unexpected and exciting features of GTPase cycling, and it introduces and characterizes new tools and approaches.

With respect to time, it is unfortunately not just "an experiment", rather it would be a minimum of 12 experiments: 6X for control groups, WT GDI expression groups, QQ GDI expression versus photoactivatable GFP-rGBD (PA-GFP-rGBD) to monitor Rho activity flux, and then 6X of each of those with PA-GFP-wGBD to monitor Cdc42 flux. The reason six replicates are required is because the photoactivation experiments are much harder than those that just require wounding since the investigator has to aim and fire a 405nm laser at the wound repeatedly as it closes. This means not only hitting the right places around the wound, but also firing at exactly the right focal plane. It takes practice, and even with practice, it is hard to get more than a few good runs in a 3-hour time slot on the microscope. In addition, the photoactivatable activity sensors express slowly in some batches of cells and very slowly in others (apparently due to slow folding of the mutant GFP). Thus, we often have to wait two days after mRNA expression for sufficient signal to conduct an experiment. This was challenging enough in the original work (Burkel et al., 2012) which took us 6 years to finish. While we are more proficient with the photoactivatable probes than when we started using them, combining them with the GDI mutants would add another level of difficulty. That is, trying to work out the details for control vs WT GDI vs QQ GDI vs. PA-GFP-rGBD expression as well as the details for control vs WT GDI vs QQ GDI vs. PA-GFP-wGBD alone would likely require several months.

8) The paper would benefit from a discussion about the complexity of spatio-temporal Rho GTPase signaling, to address the following "big picture" points: Where is the field going? What needs to be done? I would also highlight the complexity of these signaling networks, with the requirement of the zone initiation/maintenance being different for Rho and Cdc42 (as depicted in Figure 10—figure supplement 1). Further it would be good to mention that Rho GTPase -GDI interactions can be regulated by phosphorylation: both at the level of the GTPase (RhoA is phosphorylated by PKA at its C-term – this regulates GDI interaction PMID: 8599934), and at the level of GDI (by phosphorylation by PAK PMID: 15225553, or Src PMID: 16943322). These phosphorylation events might provide feedback regulation (by Rho GTPase or adhesion signaling) to spatially fine tune the system.

This is an excellent idea. We have expanded the Discussion to point out the potential impact of phosphorylation on GDI-based extraction, with a particular emphasis on phosphorylation of GDI itself and precise, local patterning of GTPases by kinase-dependent regulation of GDI working in concert with GEFs and GAPs.